# FAIRNESS-AWARE TEST-TIME PROMPT TUNING

## ABSTRACT

Vision-language models have displayed remarkable capabilities in multi-modal understanding and are increasingly used in critical applications where economic and practical deployment constraints prohibit re-training or fine-tuning. However, these models can also exhibit systematic biases that disproportionately affect protected demographic groups and existing approaches to addressing these biases require extensive model retraining and access to demographic attributes. There is a clear need to develop test-time adaptation (TTA) approaches that improve the fairness characteristics of pretrained models under distributional shift. In this paper, we evaluate how episodic TTA affects fairness in CLIP classification under subpopulation shifts and develop FAIRTPT, a novel fairness-aware episodic TTA method that jointly minimizes target marginal entropy while maximizing spurious marginal entropy through soft-prompt tuning. We find that standard episodic TTA generally exacerbates disparities between majority and minority groups, that blinding a model to spurious attributes without degrading target performance is inherently challenging, and that excessive blinding can lead to catastrophic forgetting. This model collapse can be prevented by monitoring test-time changes in target loss within the linear regime, while still achieving fairness improvements on reactive data and preserving overall performance. Thus refined, FairTPT outperforms all state-of-the-art episodic test-time debiasing methods and establishes a foundation for robust TTA—essential for achieving fairness in practice.

## 1 INTRODUCTION

Vision–language models (VLMs) such as CLIP (Radford et al., 2021) have achieved remarkable success across multi-modal tasks, including recognition, retrieval, and reasoning (Alayrac et al., 2022; Cherti et al., 2023; Li et al., 2023). A core strength of these models is zero-shot classification, which makes predictions without task-specific fine-tuning by aligning images with class-descriptive prompts in a shared embedding space. This ability is especially valuable when labeled data is scarce or fine-tuning is infeasible due to computational or deployment constraints, enabling strong cross-domain generalization (Radford et al., 2021; Jia et al., 2021). However, recent studies show that VLMs often inherit and sometimes amplify systematic biases from pre-training data (Birhane et al., 2021; Agarwal et al., 2021; Hamidieh et al., 2024; Konavoor et al., 2025; Wang et al., 2021b; Hall et al., 2023). Such biases can propagate to downstream zero-shot classifiers, leading to disparities in performance across sensitive attributes (e.g., gender, race) and undermine fairness in socially critical domains (Yang et al., 2025).

Test-time adaptation (TTA) has emerged as a promising approach to improve generalization beyond zero-shot prediction by adapting models to unseen distributions during inference using only unlabeled test inputs (Lee et al., 2022; Nath Kundu et al., 2022; Gong et al., 2022; Goyal et al., 2022; Sinha et al., 2023; Chen et al., 2022). For VLMs, episodic TTA methods such as Test-Time Prompt Tuning (TPT) (Shu et al., 2022) and ZERO (Farina et al., 2024) adapt soft prompts or aggregate predictions over augmented views to improve accuracy. While these methods improve average accuracy, their impact on subgroup robustness (i.e., the ability to maintain performance across all sensitive groups) and fairness remains largely unexplored. Existing evaluations primarily focus on overall performance, overlooking fairness-related metrics. Moreover, prior work has highlighted instability and hyperparameter sensitivity in TTA methods (Zhao et al., 2023; Sheng et al., 2025), raising concerns about their reliability in fairness-critical settings.

We address this gap by studying fairness-aware episodic TTA for VLMs, where adaptation is performed independently for each test instance and the model is reset after each episode. Formally, given an unlabeled test image $x$ with neither the target attribute $y$ nor the sensitive attribute $s$, the goal is to adapt the model (through prompt tuning) so that: (i) subgroup robustness is improved, i.e., disparities in accuracy across sensitive attributes are reduced, (ii) overall accuracy is maintained, and (iii) hyperparameter sensitivity is minimized, since tuning at test time is impractical. Existing test-time debiasing methods (Gerych et al., 2024; Adila et al., 2024; Lu et al., 2024) often require batched test inputs or partial supervision of sensitive attributes, making them unsuitable for strict, fully unsupervised episodic adaptation (cf. Table 2 in Appendix A). To our knowledge, only Chuang et al. (2023) propose an episodic unsupervised debiasing method (ORTHCALI). It projects out biased directions in text embeddings in a zero-shot setting, rather than following a standard TTA paradigm.

We introduce Fairness-Aware Test-Time Prompt Tuning (FAIRTPT), an episodic TTA method that jointly minimizes the marginal entropy of target attribute predictions to encourage overall accuracy, while maximizing the marginal entropy of sensitive attribute predictions to reduce reliance on spurious correlations. This dual-entropy objective mitigates subgroup disparities while preserving the simplicity and efficiency of prompt-level adaptation. To further stabilize adaptation and prevent catastrophic forgetting (where excessive debiasing degrades target performance), we propose a lightweight learning-rate adaptation heuristic that monitors target entropy changes during adaptation. Our method is fully unsupervised, operates in the episodic setting, and requires no access to sensitive attribute labels at test time.

Our contributions are summarized as follows:

1. **Evaluating fairness of episodic TTA methods.** We present the first systematic fairness evaluation of episodic TTA methods (TPT, ZERO) for VLMs, showing that they often fail to improve subgroup robustness, can exacerbate disparities, and are highly sensitive to hyperparameters.

2. **Proposing and evaluating a fairness-aware episodic TTA method.** We propose FAIRTPT, which jointly mitigates spurious reliance and preserves overall accuracy, while being more robust to hyperparameter choices. On standard fairness benchmarks, FAIRTPT outperforms or matches baselines (TPT, ZERO, ORTHCALI) in both overall and subgroup accuracy, and shows markedly improved stability under varying hyperparameters.

## 2 RELATED WORK

**Test-time adaptation of VLMs.** TTA methods (Wang et al., 2021a; Zhang et al., 2022; Niu et al., 2023; Shu et al., 2022; Xiao & Snoek, 2024) adapt models during inference with minimal computation, either via partial parameter tuning or tuning-free strategies (Farina et al., 2024). Popular approaches like MEMO (Zhang et al., 2022), Tent (Wang et al., 2021a), and TPT (Shu et al., 2022) leverage entropy minimization to encourage confident predictions. TTA can be applied in online settings, where the model is updated continually across test samples, or in episodic settings, where adaptation is reset to the base model for each episode or input. The latter is the focus of this work. While effective for improving average accuracy under distribution shift, TTA methods can be unstable and hyperparameter-sensitive (Zhao et al., 2023; Sheng et al., 2025), with fairness impacts largely unstudied.

**Training-time debiasing of VLMs.** Many works mitigate bias during training or fine-tuning (Wang et al., 2021b; Berg et al., 2022; Zhang & Re, 2022; Seth et al., 2023). Examples include removing embedding dimensions correlated with sensitive attributes (Wang et al., 2021b), adversarial prompt learning (Berg et al., 2022), residual feature debiasing (Seth et al., 2023), and contrastive adapter training (Zhang & Re, 2022). Broader work on group robustness includes robust optimization with group labels (Arjovsky et al., 2019; Sagawa et al., 2020) and label-free methods such as LfF (Nam et al., 2020) and JTT (Liu et al., 2021). While effective, these approaches require retraining on labeled datasets, making them impractical for deployment-time bias mitigation.

**Test-time debiasing of VLMs.** At inference, most debiasing methods perform embedding space projections to remove biased directions (Chuang et al., 2023; Gerych et al., 2024; Adila et al.,

2024; Lu et al., 2024). However, many are not fully unsupervised with respect to sensitive attributes (Gerych et al., 2024; Adila et al., 2024; Lu et al., 2024), require batched inputs (Adila et al., 2024; Lu et al., 2024), or rely on LLM-generated attribute descriptions (Adila et al., 2024; An et al., 2024). To our knowledge, ORTHCALI (Chuang et al., 2023) is the only fully episodic, unsupervised baseline applicable to our setting; however, it does not follow a standard TTA adaptation process. This gap motivates our exploration of fairness-aware, entropy-based episodic TTA for VLMs.

## 3 PROBLEM SETUP AND BACKGROUND

**Problem Setup.** We consider a test-time image classification task in a zero-shot or test-time adaptive setting, where the goal is to predict a target attribute from an input image without access to any task-specific labeled training data. Each test instance consists of an image $x \in \mathcal{X}$, an unknown target label $y \in \mathcal{Y} = \{y_1, y_2, \dots, y_C\}$, and a sensitive or spurious attribute $s \in \mathcal{S} = \{s_1, s_2, \dots, s_G\}$. The classifier only receives $x$ at inference time; $y$ and $s$ are used solely for evaluation.

Since no labeled data is available at test time, we employ zero-shot or test-time adaptive classifiers derived from vision-language models (VLMs) such as the Contrastive Language-Image Pre-training (CLIP) model (Radford et al., 2021). These models are pre-trained on large-scale image-text datasets and can perform recognition tasks directly by aligning image features with prompt-based textual features. However, due to correlations in the pretraining data, VLMs may encode spurious associations between the target label $y$ and the sensitive attribute $s$. For instance, textual descriptions of certain occupations may be disproportionately associated with one gender. Consequently, classifiers derived from such VLMs can exhibit subgroup performance disparities, particularly when the joint or marginal distributions of $(x, y, s)$ shifts between the pretraining and test domains.

To systematically quantify and analyze these disparities, we adopt the group robustness framework of Sagawa et al. (2020). Let $P_{\text{test}}$ denote the test distribution over triples $(x, y, s)$. For a classifier $f : \mathcal{X} \to \mathcal{Y}$, we define the average error as $\text{ERR}^{\text{avg}}(f) = \mathbb{E}_{(x,y) \sim P_{\text{test}}}[\mathbf{1}\{f(x) \neq y\}]$, and the worst-group error as $\text{ERR}^{\text{wg}}(f) = \max_{s \in \mathcal{S}} \mathbb{E}_{(x,y) \sim P_{\text{test}}|s}[\mathbf{1}\{f(x) \neq y\}]$, where $P_{\text{test}} \mid s$ denotes the conditional distribution given the sensitive attribute value $s$. The robustness gap is then defined as the difference $\text{GAP}(f) = \text{ERR}^{\text{avg}}(f) - \text{ERR}^{\text{wg}}(f)$, capturing the discrepancy between the overall and worst-case group performance. The goal of this work is to design test-time adaptive classifiers that achieve high overall accuracy while minimizing the robustness gap, thereby promoting fairness under distribution shifts along sensitive attributes.

**Zero-Shot Classification.** To perform zero-shot classification, we leverage a pre-trained VLM, which learns a shared embedding space for images and natural language prompts. The model consists of an image encoder $f_{\text{V}}$, which maps an image $x \in \mathcal{X}$ to a feature vector $\boldsymbol{v} = f_{\text{V}}(x) \in \mathbb{R}^d$, and a text encoder $f_{\text{L}}$, which maps a text sequence (e.g., a class-descriptive prompt) to a feature vector $\boldsymbol{l} \in \mathbb{R}^d$. The classifier predicts the target attribute $y \in \mathcal{Y}$ by measuring the similarity between the image feature $\boldsymbol{v}$ and a set of class-specific text features $\boldsymbol{l}_y$ derived from prompts.

To construct these prompts, a hand-crafted template containing a placeholder (e.g., `"a photo of a [T-CLS] person"` where `[T-CLS]` is the placeholder for a target class) is instantiated with a textual description for each target label $y \in \mathcal{Y}$. For example, for the attribute `"smiling"`, we may use: { `"a photo of a smiling person"`, `"a photo of a non-smiling person"` }. These completed prompts are tokenized and embedded into soft prompt representations using the model's token embedding table. For simplicity, each soft prompt is represented as a non-ordered tuple $(\boldsymbol{t}_{\text{ctx}}, \boldsymbol{t}_y) \in \mathbb{R}^{L_y \times d}$, where $\boldsymbol{t}_{\text{ctx}}$ encodes the shared context tokens, and $\boldsymbol{t}_y$ encodes the label-specific portion. The length $L_y$ corresponds to the total number of tokens in the prompt for label $y$, and $d$ is the token embedding dimension. This results in a soft prompt set $\{\boldsymbol{t}_{\text{ctx}}; \mathcal{Y}\} = \{(\boldsymbol{t}_{\text{ctx}}, \boldsymbol{t}_y) : y \in \mathcal{Y}\}$, which is used to generate the set of text features $\{\boldsymbol{l}_y : y \in \mathcal{Y}\}$, where $\boldsymbol{l}_y = f_{\text{L}}((\boldsymbol{t}_{\text{ctx}}, \boldsymbol{t}_y))$.

During inference, the image feature $\boldsymbol{v}$ is compared to each text feature $\boldsymbol{l}_y$ using cosine similarity, denoted $\text{sim}(\boldsymbol{l}_y, \boldsymbol{v}) = \frac{\langle \boldsymbol{l}_y, \boldsymbol{v} \rangle}{\|\boldsymbol{l}_y\| \cdot \|\boldsymbol{v}\|}$. These similarity scores are passed through a softmax function to produce a probability distribution over the target classes:

$$p(y \mid x, \{\boldsymbol{t}_{\text{ctx}}; \mathcal{Y}\}, \tau) = \frac{\exp\left(\tau \cdot \text{sim}(\boldsymbol{l}_y, \boldsymbol{v})\right)}{\sum_{y' \in \mathcal{Y}} \exp\left(\tau \cdot \text{sim}(\boldsymbol{l}_{y'}, \boldsymbol{v})\right)}, \quad \forall y \in \mathcal{Y},$$

where $\tau > 0$ is a temperature parameter that controls the sharpness of the output distribution. The final prediction is chosen from the above distribution: $\widehat{y} = \mathrm{argmax}_y p(y \mid x, \{\boldsymbol{t}_{\mathrm{ctx}}; \mathcal{Y}\}, \tau)$. This zero-shot framework forms the foundation upon which we develop our fairness-aware prompt tuning method.

**Test-Time Adaptation (TTA).** We study episodic test-time adaptation, where the model processes one test image at a time and is allowed to adapt its parameters within a reasonable time before producing the final prediction. The model receives an image $x \in \mathcal{X}$ and has no access to its true label $y$ or sensitive attribute $s$ during inference. Adaptation is performed using a pre-trained VLM $\{f_{\mathrm{V}}, f_{\mathrm{L}}\}$, together with a set of augmentation functions $\mathcal{A} = \{A_1, \ldots, A_N\}$ that generate alternative views of $x$. This is performed using AugMix (Hendrycks et al., 2020). The aim is to leverage these augmented views to improve prediction robustness under distribution shifts.

We first consider Test-Time Prompt Tuning (TPT) (Shu et al., 2022), an adaptation of marginal entropy minimization (MEM) (Zhang et al., 2022) to the vision-language setting. MEM was originally proposed for unimodal vision models, but Shu et al. (2022) demonstrated that it can be repurposed for VLMs by optimizing prompt embeddings rather than the model weights (Zhou et al., 2022b;a; Khattak et al., 2023). Specifically, a trainable soft-prompt set $\{\boldsymbol{t}_{\mathrm{ctx}}; \mathcal{Y}\} = \{(\boldsymbol{t}_{\mathrm{ctx}}, \boldsymbol{t}_y) : y \in \mathcal{Y}\}$ is constructed as in the zero-shot procedure, where $\boldsymbol{t}_{\mathrm{ctx}}$ represents the shared context and $\boldsymbol{t}_y$ encodes the label-specific portion. For each test image $x$, the augmentation set $\mathcal{A}$ is applied to generate $N$ views, producing a marginal predictive distribution:

$$\overline{p}(y \mid x, \{\boldsymbol{t}_{\mathrm{ctx}}; \mathcal{Y}\}, \tau) = \frac{1}{N} \sum_{i=1}^{N} p(y \mid A_i(x), \{\boldsymbol{t}_{\mathrm{ctx}}; \mathcal{Y}\}, \tau), \quad \forall y \in \mathcal{Y},$$

where $p(\cdot \mid \cdot)$ is computed as in the zero-shot classifier and $\tau > 0$ is the softmax temperature. Justified by the desired augmentation-invariance and improved confidence of the predictions (Zhang et al., 2022), the adaptation objective is to minimize the entropy of this marginal distribution for each input $x$:

$$\min_{\boldsymbol{t}_{\mathrm{ctx}}} H(\overline{p}(\cdot \mid x, \{\boldsymbol{t}_{\mathrm{ctx}}; \mathcal{Y}\}, \tau)) := -\sum_{y \in \mathcal{Y}} \overline{p}(y \mid x, \{\boldsymbol{t}_{\mathrm{ctx}}; \mathcal{Y}\}, \tau) \cdot \log \overline{p}(y \mid x, \{\boldsymbol{t}_{\mathrm{ctx}}; \mathcal{Y}\}, \tau). \quad (1)$$

This optimization is performed for $n_{\mathrm{epochs}}$ iterations using an update rule $G$ with learning rate $\eta$, yielding an adapted context embedding $\boldsymbol{t}_{\mathrm{ctx}}^*$. The final prediction is then obtained by applying the zero-shot classification procedure with the updated soft-prompt set $\{\boldsymbol{t}_{\mathrm{ctx}}^*; \mathcal{Y}\}$. Following Shu et al. (2022), we employ AdamW (Loshchilov & Hutter, 2019) as the optimizer, with $\eta = 5e - 3$ and $n_{\mathrm{epochs}} = 1$.[1]

As a recent and simpler alternative, we also consider the ZERO method of Farina et al. (2024), which avoids any adaptation step and instead aggregates predictions from the augmented views directly. The predicted label is obtained as

$$\mathrm{ZERO}(x, \{\boldsymbol{t}_{\mathrm{ctx}}; \mathcal{Y}\}) = \arg\max_{y \in \mathcal{Y}} \sum_{i=1}^{N} \lim_{\tau \to 0^+} p(y \mid A_i(x), \{\boldsymbol{t}_{\mathrm{ctx}}; \mathcal{Y}\}, \tau), \quad (2)$$

corresponding to majority voting over deterministic predictions from each view.

For both TPT and ZERO, performance and stability can be improved through confidence-based view filtering (Shu et al., 2022; Niu et al., 2023; Farina et al., 2024). The idea is to retain the top-$\rho$ fraction of high-confidence (low-entropy) augmented views for the target attribute prediction. High-entropy views are discarded as they often lack sufficient information for reliable classification. Formally, the retained augmentation set is $\mathcal{A}_{\mathrm{filtered}}(x) = \{A_i : H(p(\cdot \mid A_i(x), \{\boldsymbol{t}_{\mathrm{ctx}}; \mathcal{Y}\}, \tau)) < \delta\}$, where $\delta$ is a threshold chosen to retain the top-$\rho$ fraction of most confident views. In both cases, the original studies were setting $\rho = 0.1$.

## 4 FAIRNESS-AWARE TPT

**Fairness-Aware Test-Time Prompt Tuning (FAIRTPT).** We propose Fairness-Aware Test-Time Prompt Tuning (FAIRTPT), a fairness-aware extension of test-time prompt tuning designed to im-

---

[1]Note that we did not use CoOp or CoCoOp (Zhou et al., 2022b;a), as they were not necessary to achieve a performance uplift in the original study.

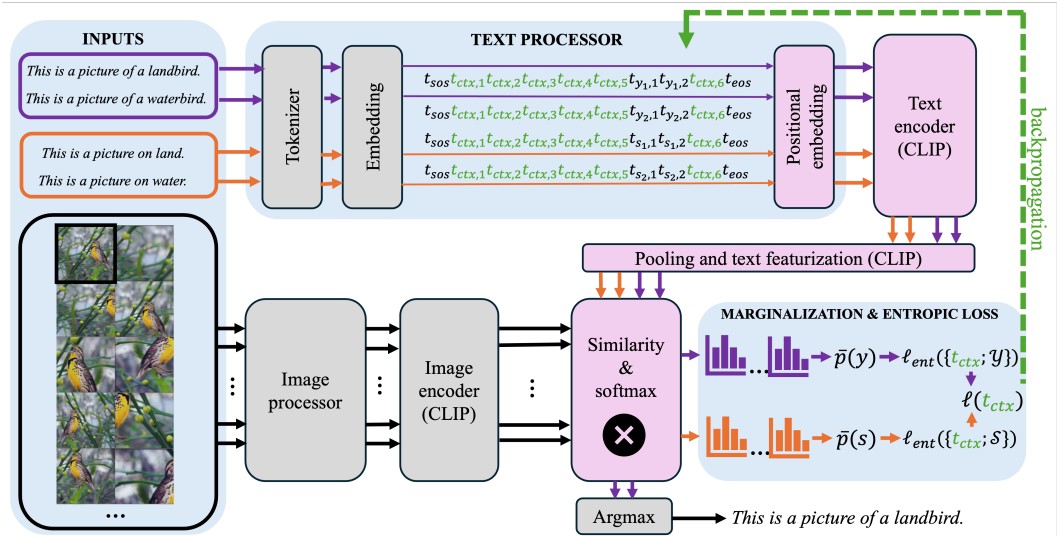

Figure 1: FAIRTPT pipeline using a single unlabeled image at a time (black frame). The soft prompt $t_{ctx}$ is the only parameter tuned at test time, requiring text processor to be re-written. Pink and grey blocks have frozen weights; gradients are computed only for pink blocks. This example illustrates the objective in Equation 3. For the alternative objective in Equation 4, the spurious prompts (orange) and their corresponding soft prompts would also include target labels, resulting in four spurious inputs in this example.

prove subgroup robustness while preserving overall accuracy. As in the zero-shot classification setting, we construct a trainable target soft-prompt set $\{t_{ctx}; \mathcal{Y}\} = \{(t_{ctx}, t_y) : y \in \mathcal{Y}\}$ where $t_{ctx}$ is the shared context embedding and $t_y$ encodes the target label description. In parallel, we construct a spurious soft-prompt set $\{t_{ctx}; \mathcal{S}\} = \{(t_{ctx}, t_s) : s \in \mathcal{S}\}$ for the sensitive attribute, reusing the same shared context $t_{ctx}$ across both sets.

Following the marginal entropy minimization framework in TPT, we compute two marginal predictive distributions: the target marginal $\overline{p}(\cdot \mid x, \{t_{ctx}; \mathcal{Y}\}, \tau)$ over $\mathcal{Y}$ and the spurious marginal $\overline{p}(\cdot \mid x, \{t_{ctx}; \mathcal{S}\}, \tau)$ over $\mathcal{S}$, each obtained by averaging predictions over the same augmented views $\{A_i(x) : A_i \in \mathcal{A}\}$. To jointly promote confident predictions for the target attribute and uncertainty for the spurious attribute, we optimize the objective

$$\min_{t_{ctx}} \frac{1}{1 + \lambda_{\text{fair}}} \ell_{\text{ent}}(x, \{t_{ctx}; \mathcal{Y}\}) - \frac{\lambda_{\text{fair}}}{1 + \lambda_{\text{fair}}} \cdot \ell_{\text{ent}}(x, \{t_{ctx}; \mathcal{S}\}), \tag{3}$$

where $\lambda_{\text{fair}} \geq 0$ balances the accuracy–fairness trade-off[2], and

$$\ell_{\text{ent}}(x, \{t_{ctx}; \star\}) = \frac{1}{\log |\star|} \cdot H(\overline{p}(\cdot \mid x, \{t_{ctx}; \star\}, \tau)), \quad \text{for } \star \in \{\mathcal{Y}, \mathcal{S}\}.$$

The objective is minimized for $n_{\text{epochs}}$ iterations using a gradient-based update rule $G$ with learning rate $\eta$, producing an adapted context embedding $t_{ctx}^*$. The final prediction is then obtained by applying the zero-shot classifier with the updated target soft-prompt set $\{t_{ctx}^*; \mathcal{Y}\}$. The complete FAIRTPT procedure is provided in Algorithm 3 (see Appendix A.1).

**Alternative Formulation for the Test-Time Optimization.** We also consider an alternative formulation in which spurious prompts are conditioned on target label information. In this case, we define a joint prompt template containing two placeholders, one for the spurious attribute and one for the target attribute, e.g., `"a photo of a [S-CLS] celebrity [T-CLS]"`. For each pair $(y, s) \in \mathcal{Y} \times \mathcal{S}$, this yields a prompt such as `"a photo of a male celebrity`

---
[2]Note that $\lambda_{\text{fair}} = 0$ recovers the TPT baseline, and $\lambda_{\text{fair}} \to \infty$ corresponds to optimizing solely for reduced confidence in the spurious attribute, ignoring the target attribute.

smiling" or "a photo of a female celebrity smiling". Each prompt is embedded as $(\boldsymbol{t}_{\text{ctx}}, \boldsymbol{t}_y, \boldsymbol{t}_s) \in \mathbb{R}^{L_{ys} \times d}$, and for a fixed $y$ we construct a spurious soft-prompt set $\{\boldsymbol{t}_{\text{ctx}}, \boldsymbol{t}_y; \mathcal{S}\} = \{(\boldsymbol{t}_{\text{ctx}}, \boldsymbol{t}_y, \boldsymbol{t}_s) : s \in \mathcal{S}\}$. The corresponding marginal distribution $\overline{p}(\cdot \mid x, \{\boldsymbol{t}_{\text{ctx}}, \boldsymbol{t}_y; \mathcal{S}\}, \tau)$ is then used to replace the spurious term in the objective in Equation 3:

$$\min_{\boldsymbol{t}_{\text{ctx}}} \frac{1}{1 + \lambda_{\text{fair}}} \ell_{\text{ent}}(x, \{\boldsymbol{t}_{\text{ctx}}; \mathcal{Y}\}) - \frac{\lambda_{\text{fair}}}{1 + \lambda_{\text{fair}}} \cdot \frac{1}{|\mathcal{Y}|} \sum_{y \in \mathcal{Y}} \ell_{\text{ent}}(x, \{\boldsymbol{t}_{\text{ctx}}, \boldsymbol{t}_y; \mathcal{S}\}), \tag{4}$$

where

$$\ell_{\text{ent}}(x, \{\boldsymbol{t}_{\text{ctx}}, \boldsymbol{t}_y; \mathcal{S}\}) = \frac{1}{\log |\mathcal{S}|} \cdot H(\overline{p}(\cdot \mid x, \{\boldsymbol{t}_{\text{ctx}}, \boldsymbol{t}_y; \mathcal{S}\}, \tau)).$$

We refer to the objective in Equation 3 as the *S* loss, and the objective in Equation 4 as the *TS* loss.

It is also possible to make the context $\boldsymbol{t}_{\text{ctx}}$ target-dependent and optimize it separately for each target soft prompt:

$$\min_{\{\boldsymbol{t}_{\text{ctx}}^{(y)}, y \in \mathcal{Y}\}} \frac{1}{1 + \lambda_{\text{fair}}} \cdot \frac{1}{|\mathcal{Y}|} \sum_{y \in \mathcal{Y}} \ell_{\text{ent}}(x, \{\boldsymbol{t}_{\text{ctx}}^{(y)}; \mathcal{Y}\}) - \frac{\lambda_{\text{fair}}}{1 + \lambda_{\text{fair}}} \cdot \frac{1}{|\mathcal{Y}|} \sum_{y \in \mathcal{Y}} \ell_{\text{ent}}(x, \{\boldsymbol{t}_{\text{ctx}}^{(y)}, \boldsymbol{t}_y; \mathcal{S}\}), \tag{5}$$

where $\boldsymbol{t}_{\text{ctx}}^{(y)}$ is the tunable soft context for target $y$ and excludes its tokens. We refer to this second alternative as the *Super TS* loss.

**Learning Rate Adaptation.** Large gradient updates during test-time adaptation can degrade performance, sometimes even dropping below that of zero-shot classification (Niu et al., 2023). To mitigate the risk of model collapse and monitor the adaptation speed, we employ a learning rate adaptation heuristic, whose implementation is described below and presented in Algorithm 4.

We choose the initial learning rate $\eta_{\text{init}}$ from a regime in which $\boldsymbol{t}_{\text{ctx}}$ undergoes only a small change in norm after a single optimization step. This defines the linear regime, where a first-order Taylor expansion of the loss function $\ell$ with respect to $\boldsymbol{t}_{\text{ctx}}$ is valid. For a simple optimizer such as SGD, the update rule is $G : \boldsymbol{t}_{\text{ctx}} \leftarrow \boldsymbol{t}_{\text{ctx}} - \eta \cdot \nabla_{\boldsymbol{t}_{\text{ctx}}} \ell$. The linear regime is achieved when $\eta_{\text{init}} = \sigma \cdot \|\boldsymbol{t}_{\text{ctx}}\| / \|\nabla_{\boldsymbol{t}_{\text{ctx}}} \ell\|$ for $\sigma \ll 1$, where we set $\sigma = 0.01$ in our pipeline.

Operating in this regime allows direct control over parameter changes by linearly scaling the learning rate. After each optimization step, we monitor the absolute change in the target loss, $|\Delta \ell_{\text{ent}}(x, \{\boldsymbol{t}_{\text{ctx}}; \mathcal{Y}\})|$, and bound it by $\beta$ via the rescaling $\eta \leftarrow \eta \cdot \beta / |\Delta \ell_{\text{ent}}(x, \{\boldsymbol{t}_{\text{ctx}}; \mathcal{Y}\})|$. This dynamic adjustment stabilizes adaptation and prevents divergence from overly large updates. Moreover, continuity of the pre-argmax model implies that bounding parameter changes also bounds logit variations; hence, small parameter updates in the linear regime yield predictable and controlled accuracy changes.

**Multi-Objective Optimization.** The objective in Equation 3 (or Equation 4) comprises the target term $\ell_{\text{ent}}(x, \{\boldsymbol{t}_{\text{ctx}}; \mathcal{Y}\})$ and the spurious term $-\ell_{\text{ent}}(x, \{\boldsymbol{t}_{\text{ctx}}; \mathcal{S}\})$, the gradients of which may conflict. Optimizing only their combined scalar loss risks improving one term at the expense of the other. We therefore cast the problem as a multi-objective optimization (MO) task:

$$\min_{\boldsymbol{t}_{\text{ctx}}} (\ell_{\text{ent}}(x, \{\boldsymbol{t}_{\text{ctx}}; \mathcal{Y}\}), -\ell_{\text{ent}}(x, \{\boldsymbol{t}_{\text{ctx}}; \mathcal{S}\})). \tag{6}$$

We solve this MO problem using Jacobian descent (Quinton & Rey, 2024), which optimizes vector-valued objectives by computing per-term gradients and aggregating them via a chosen aggregator. The aggregator can incorporate a weighting parameter $\lambda_{\text{fair (mo)}}$ for the spurious term. Jacobian descent guarantees a simultaneous decrease in both objectives, provided the learning rate is sufficiently small (Quinton & Rey, 2024). In our experiments, we use the TorchJD library[3] with the Unconflicting Projection of Gradients (UPGrad) aggregator.

## 5 EXPERIMENTS

In this section, we comprehensively evaluate FAIRTPT against episodic test-time adaptation (TTA) baselines (TPT and ZERO) as well as an episodic test-time debiasing baseline, ORTHCALI. Ablations are also performed to quantify the impact of key design choices and hyperparameters.

---

[3] https://torchjd.org/stable/

## 5.1 EXPERIMENTAL SETUP

**Base Model.** We use CLIP (Radford et al., 2021), specifically the `ViT-L/14` variant (approximately 428M parameters), obtained from Hugging Face.[4]

**Datasets.** We evaluate on standard benchmarks from the algorithmic fairness literature: CELEBA (Liu et al., 2015), UTKFACE (Zhang et al., 2017), FAIRFACE (Karkkainen & Joo, 2021), and WATERBIRDS (Sagawa et al., 2020). For each run and each dataset, we uniformly sample $K = 1000$ images. The target and spurious attributes used in our experiments are specified in Table 5 (Appendix A.2), and the initial prompt templates with placeholders for target and sensitive attributes, which are required by FAIRTPT and the episodic TTA baselines, are given in Table 4 (Appendix A.2). Across these datasets, zero-shot accuracy of the base model ranges from roughly 70% to over 95%, while the corresponding bias (formally defined below) ranges from about 5% to over 40%. These characteristics make the suite well-suited for probing the behavior of test-time adaptation and debiasing methods under subpopulation distribution shift.

**Evaluation Metrics.** Let $M \in \mathbb{R}^{C \times C}$ denote the overall confusion matrix, where $M_{ij}$ counts examples with true label $y_i$ predicted as $y_j$ across $K$ test episodes, and let $M^{(s)} \in \mathbb{R}^{C \times C}$ be the confusion matrix restricted to group $s \in \mathcal{S}$ with $K^{(s)}$ episodes. We report overall accuracy ACC $= \frac{1}{K} \sum_{i \in [C]} M_{ii}$, worst-group accuracy WGA $= \min_{s \in \mathcal{S}} \frac{1}{K^{(s)}} \sum_{i \in [C]} M_{ii}^{(s)}$, and bias BIAS $=$ ACC $-$ WGA. We also measure equalized-odds difference, defined as EOD $= \operatorname{mean}_{y_i} \max\{\operatorname{Gap}_s \operatorname{TPR}_{y_i}^{(s)}, \operatorname{Gap}_s \operatorname{FPR}_{y_i}^{(s)}\}$, where $\operatorname{TPR}_{y_i}^{(s)} = M_{ii}^{(s)} / \sum_{j \in [C]} M_{ij}^{(s)}$, $\operatorname{FPR}_{y_i}^{(s)} = \sum_{j \in [C], j \neq i} M_{ji}^{(s)} / \sum_{j \in [C], j \neq i} \sum_{k \in [C]} M_{jk}^{(s)}$, and $\operatorname{Gap}_s \operatorname{TPR}_{y_i}^{(s)} = \max_{s \in \mathcal{S}} \operatorname{TPR}_{y_i}^{(s)} - \min_{s \in \mathcal{S}} \operatorname{TPR}_{y_i}^{(s)}$ (analogously for FPR).

## 5.2 RESULTS AND DISCUSSION

All results are averaged over five independent runs with different random seeds. We report (i) overall performance using overall accuracy, and (ii) subgroup-level performance using worst-group accuracy (WGA; higher is better), bias (lower is better), and equalized odds difference (EOD; lower is better). Unless otherwise stated, "our method" refers to both FAIRTPT and FAIRTPT (MO), each equipped with ELRA and $S$ loss.

**Questions and evaluation protocol.** We organize the analysis around three evaluations. **E1** probes the behavior of episodic TTA methods designed to improve overall accuracy (TPT and ZERO), including their fairness implications and hyperparameter sensitivity. **E2** examines an episodic test-time debiasing baseline (ORTHCALI), focusing on whether it improves subgroup performance without sacrificing overall accuracy, and how sensitive it is to its hyperparameters. **E3** evaluates FAIRTPT (and FAIRTPT (MO)), asking whether it retains or improves overall and subgroup-level performance relative to zero-shot and baselines, how sensitive it is to hyperparameters, and which components matter through ablations. We report both aggregated trends (averaged across datasets) and per-dataset behavior.

**Main comparison.** Table 1 summarizes the overall and subgroup-level results. For all baselines we keep their recommended hyperparameters fixed across datasets. See Table 3 (Appendix A.1) for the hyperparameter values of our methods and baselines. Overall, when averaged across all datasets, our methods' accuracy and subgroup metrics are comparable to or better than those of all baselines.

**Overall accuracy (E1-3).** Episodic TTA methods do not consistently improve accuracy over zero-shot across datasets. We observe cases with improvements exceeding +2.0pp (percentage points) and others with degradations below -2.0pp; on aggregate, accuracy remains close to or marginally below zero-shot. This aligns with recent reports on the brittleness of episodic TTA methods, which have rarely been benchmarked on fairness-oriented datasets. Even when incorporating model-recovery and sharpness-aware updates (Niu et al., 2023), we did not see consistent gains. ORTHCALI generally retains zero-shot accuracy, with the exception of CELEBA (Hair color × Gender), where

---

[4]`https://huggingface.co/openai/clip-vit-large-patch14`

| METHOD | FAIRFACE | | | | CELEBA | | | | | | | | WATERBIRDS | | | | UTKFACE | | | | | | | | AVERAGE RESULTS | | | |
|---|---|---|---|---|---|---|---|---|---|---|---|---|---|---|---|---|---|---|---|---|---|---|---|---|---|---|---|---|
| | Gender × Race | | | | Hair color × Gender | | | | Smiling × Gender | | | | Type × Background | | | | Age × Race | | | | Gender × Race | | | | | | | |
| | A | WGA | B | EOD | A | WGA | B | EOD | A | WGA | B | EOD | A | WGA | B | EOD | A | WGA | B | EOD | A | WGA | B | EOD | A | WGA | B | EOD |
| ZERO-SHOT | 95.7 | 90.0 | 5.7 | 9.4 | 86.4 | 67.8 | 18.6 | 22.3 | 75.8 | 53.7 | 22.1 | 8.0 | 83.8 | 40.2 | 43.7 | 25.0 | 80.3 | 45.7 | 34.5 | 23.2 | 97.1 | 90.1 | 7.0 | 9.1 | 86.5 | 64.6 | 21.9 | 16.2 |
| *Episodic test-time adaptation methods* | | | | | | | | | | | | | | | | | | | | | | | | | | | | |
| TPT | 93.6 | 85.0 | 8.6 | 12.4 | 92.0 | 42.6 | 49.4 | 37.6 | 59.4 | 20.0 | 39.4 | 5.0 | 84.0 | 40.8 | 43.1 | 37.1 | 87.5 | 57.7 | 29.8 | 26.3 | 94.3 | 86.4 | 7.9 | 8.0 | 85.1 | 55.4 | 29.7 | 21.1 |
| ZERO | 91.2 | 78.6 | 12.6 | 11.9 | 90.4 | 56.3 | 34.0 | 23.8 | 69.9 | 38.0 | 31.9 | 11.1 | 83.1 | 40.0 | 43.2 | 29.0 | 86.2 | 53.8 | 32.4 | 28.7 | 93.5 | 77.8 | 15.7 | 13.4 | 85.7 | 57.4 | 28.3 | 19.6 |
| *Episodic test-time debiasing methods* | | | | | | | | | | | | | | | | | | | | | | | | | | | | |
| ORTHCALI | 95.9 | 87.8 | 8.1 | 11.3 | 85.1 | 71.6 | 13.5 | 22.0 | 71.4 | 36.8 | 34.6 | 30.6 | 83.3 | 62.0 | 21.4 | 16.2 | 79.8 | 47.0 | 32.8 | 14.4 | 96.8 | 88.9 | 7.9 | 10.1 | 85.4 | 65.7 | 19.7 | 17.4 |
| *Our method* | | | | | | | | | | | | | | | | | | | | | | | | | | | | |
| FAIRTPT | 95.5 | 90.7 | 4.8 | 8.1 | 85.6 | 67.7 | 17.9 | 21.9 | 75.9 | 56.1 | 19.8 | 6.2 | 83.2 | 42.4 | 40.8 | 23.5 | 81.0 | 51.5 | 29.5 | 23.4 | 96.7 | 90.6 | 6.1 | 8.6 | 86.3 | 66.5 | 19.8 | 15.3 |
| FAIRTPT (MO) | 95.3 | 90.4 | 4.9 | 7.9 | 85.3 | 67.5 | 17.8 | 21.3 | 76.1 | 57.8 | 18.3 | 6.3 | 83.2 | 41.4 | 41.8 | 24.6 | 80.8 | 51.6 | 29.3 | 20.9 | 96.6 | 90.9 | 5.8 | 8.1 | 86.2 | 66.6 | 19.6 | 14.8 |

Table 1: Overall (Accuracy) and subgroup-level performance (Worst-Group Accuracy, Bias, and Equalized Odds Difference) evaluation of all the methods considered (ours and baselines). We report mean over 5 random seeds and mean aggregation over all equally sized datasets. Best results are in **bold**, second best underlined. The values that improve upon ZERO-SHOT by more than 2.0 percentage points are highlighted in green, while degradations greater than 2.0 percentage points are shown in red (arbitrary threshold). We set $\lambda_{\text{fair}} = 100$ and $\lambda_{\text{fair (mo)}} = 100$ for FAIRTPT and FAIRTPT (MO), respectively. Hyperparameter values of all methods are presented in Table 3 (Appendix A.1).

it underperforms. On aggregate, accuracy for ORTHCALI remains close to or marginally below zero-shot. Both FAIRTPT variants consistently retain zero-shot accuracy across datasets, and their aggregated accuracy is on par with (or marginally below) zero-shot.

**Subgroup metrics (E1-3).** In most dataset/attribute configurations, episodic TTA methods worsen subgroup performance relative to zero-shot, with more than -2.0pp (percentage points) drops observed in WGA, Bias, and EOD. Aggregated over datasets, all subgroup metrics degrade by more than -2.0pp. This underscores fairness risks of episodic TTA under subpopulation shift. ORTHCALI does not improve subgroup metrics consistently. We observe both improvements beyond +2.0pp and degradations below -2.0pp depending on the dataset; on aggregate, WGA and Bias slightly improve while EOD slightly degrades. FAIRTPT and FAIRTPT (MO) retain or improve subgroup metrics across datasets, with several cases exceeding +2.0pp gains. Aggregated over datasets, all subgroup metrics improve, with WGA and Bias improving by more than +2.0pp.

**Sensitivity of episodic TTA (E1).** We study TPT sensitivity to the learning rate $\eta$ and confidence threshold $\rho$ in Table 6 (Appendix A.3), keeping other hyperparameters fixed as in Table 3.

Effect of $\eta$. Aggregated results show that subgroup metrics (WGA, Bias, EOD) begin to degrade even at moderate $\eta \approx 1e-3$ (often beyond -2.0pp), while overall accuracy remains stable until much larger $\eta$, typically degrading around $\eta \approx 1e-1$. This trend is consistent across datasets: a broad $\eta$ range preserves accuracy, but only a narrow $\eta$ range preserves subgroup performance. The precise onset of degradation is dataset dependent. Subgroup metrics are thus more sensitive than overall accuracy to $\eta$, reinforcing the need for careful learning-rate selection at test time.

Effect of $\rho$. For fixed $\eta$, varying $\rho$ has a mild effect on aggregated performance. Per-dataset, when $\eta$ lies in a stable regime (no evident degradation), both accuracy and subgroup metrics vary minimally with $\rho$. This suggests $\rho$ is secondary relative to $\eta$ for TPT stability.

For ZERO, Table 7 (Appendix A.3) shows $\rho$ has a mild aggregate effect, with accuracy being consistently robust and subgroup metrics occasionally sensitive on certain datasets. Overall, episodic TTA methods exhibit limited robustness of subgroup performance to their key hyperparameters, even when accuracy appears stable.

**Sensitivity of ORTHCALI (E2).** Table 8 (Appendix A.3) scans $\lambda_{\text{orth}}$. Aggregated results show that all metrics except EOD are largely robust to $\lambda_{\text{orth}}$, but per-dataset analysis reveals that either accuracy or subgroup metrics begin to degrade beyond dataset-specific $\lambda_{\text{orth}}$ ranges. The $\lambda_{\text{orth}}$ values that best balance accuracy and subgroup metrics vary with the dataset and attribute pairing, indicating limited cross-dataset robustness.

**Sensitivity and ablations for FAIRTPT (E3).** Table 9 (Appendix A.3) scans $\lambda_{\text{fair}}$ (and $\lambda_{\text{fair (mo)}}$) with other settings fixed as in Table 3: $n_{\text{epochs}} = 1$ (as in TPT), $\rho = 0.75$ (chosen because TPT is

relatively insensitive to $\rho$ and a higher $\rho$ allows to maximize spurious-attribute entropy over more augmentations), and $\beta = 0.01$ (intuitive choice).

Robustness to $\lambda_{\text{fair}}$. Aggregated and per-dataset trends show that FAIRTPT and FAIRTPT (MO) retain or improve both accuracy and subgroup metrics across a wide range of $\lambda_{\text{fair}}$ and $\lambda_{\text{fair (mo)}}$ values. Any sufficiently large $\lambda_{\text{fair}}$ (from 100 up to effectively $\infty$) balances accuracy and subgroup objectives reliably across datasets.

Role of ELRA. Removing ELRA (and using AdamW with $\eta = 5e-3$ for FAIRTPT and FAIRTPT (MO)) leads to notable degradations in both accuracy and subgroup metrics on average, and similarly on most individual datasets. While careful manual tuning of $\eta$ could partially mitigate this (as suggested by the TPT sensitivity study), such tuning is impractical at test time. In contrast, ELRA adaptively selects learning rates from unlabeled inputs and interacts favorably with $\lambda_{\text{fair}}$ and $\rho$, enabling FAIRTPT variants to improve subgroup performance where possible while preventing accuracy collapse.

Variants of the spurious-entropy term. Table 10 (Appendix A.3) compares alternative formulations of the spurious-entropy component; we do not observe meaningful gains over the original formulation.

Combination of FAIRTPT and ORTHCALI. FAIRTPT is compatible with ORTHCALI. For completeness, Table 11 (Appendix A.3) provides the performance obtained by applying FAIRTPT followed by the orthogonal projection of ORTHCALI (line 6 of Algorithm 3, Appendix A.1). No collapse is observed as opposed to ORTHCALI alone but no improvement is found over FAIRTPT.

**Takeaways.** Across datasets, episodic TTA baselines are not reliably accuracy-improving and often harm subgroup performance unless hyperparameters are carefully chosen, something that is particularly hard at test time with only unlabeled inputs. ORTHCALI can retain accuracy on average and sometimes improves subgroup metrics, but the best $\lambda_{\text{orth}}$ varies by dataset, limiting plug-and-play robustness. In contrast, FAIRTPT and FAIRTPT (MO) deliver (i) subgroup improvements with (ii) retained accuracy and (iii) markedly improved robustness to hyperparameters, thus meeting the desirable criteria for fairness-aware test-time methods. Practically, using sufficiently large $\lambda_{\text{fair}}$ and a small $\beta$ yields strong and stable performance across datasets without per-dataset tuning. The robustness of these conclusions is further supported by additional experiments on other attributes, reported in Table 12 (Appendix A.3). Further details, including effectiveness validation, runtime comparison, and multi-attribute support, are provided in Appendix A.4.

# 6 CONCLUSIONS

In this work, we conducted the first fairness evaluation of episodic test-time adaptation (TTA) methods for vision–language models (VLMs) in zero-shot classification. Our analysis revealed that existing methods often fail to improve subgroup robustness, can amplify disparities, and are highly sensitive to hyperparameters. To address these shortcomings, we introduced FAIRTPT, a fully unsupervised, label-free method that jointly minimizes target-attribute entropy to preserve accuracy and maximizes sensitive-attribute entropy to reduce spurious correlations. Combined with a lightweight learning-rate adaptation heuristic, FAIRTPT mitigates collapse, stabilizes adaptation, and achieves state-of-the-art or competitive performance on fairness benchmarks, thus improving both overall and subgroup accuracy while exhibiting markedly greater robustness to hyperparameter variation.

While our results demonstrate the promise of fairness-aware episodic TTA, they also highlight open challenges and opportunities for future research. First, extending fairness-aware debiasing from the episodic to the online TTA setting could enable models to adapt continuously while maintaining subgroup robustness. Second, ensuring that confidence estimates are well-aligned for both majority and minority groups by improving subgroup-level calibration, remains an important but underexplored objective. Third, a deeper theoretical understanding of entropy-based test-time debiasing is needed to explain when and why marginal entropy maximization succeeds or fails, potentially guiding adaptive strategies that avoid over-debiasing or catastrophic forgetting. Finally, exploring meta-learned or hybrid objectives that balance accuracy, fairness, and stability could further advance the deployment of fairness-aware TTA in high-stakes, real-world applications. By bridging fairness and adaptation, we take a step toward test-time learning systems that are not only accurate under distribution shift but also equitable, stable, and trustworthy in socially critical settings.

## REPRODUCIBILITY STATEMENT

All datasets used are public. The code will be made publicly available after the anonymity period, subject to legal approval. All the hyperparameter used are presented in Table 3.

## LLM USAGE STATEMENT

A large language model (LLM) was used solely to polish the writing of this manuscript. All substantive ideas, research design, analysis, and conclusions were developed entirely by the human authors.

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

# A ADDITIONAL DETAILS AND RESULTS

## A.1 METHODS

A comparison of existing test-time debiasing methods is presented in Table 2.

| Method | Explicit debiasing | Unsupervised | Episodic | Difference from our setup |
|---|---|---|---|---|
| ORTHCALI | ✓ | ✓ | ✓ | — |
| BENDVLM | ✓ | × | × | Requires a reference dataset of images with spurious labels. |
| ROBOSHOT | × | ✓ | ✓ | Spurious attributes suggested by an LLM. |
| TIE | ✓ | × | × | Supervised with respect to the spurious attribute for each batch. |
| TIE* | ✓ | ✓ | × | Scale coefficient computed as the batch average. |
| PerceptionCLIP | × | ✓ | ✓ | Conditioning over a mix of concepts rather than targeted debiasing. |

Table 2: Comparison of state-of-the-art test-time debiasing methods for VLMs (Chuang et al., 2023; Gerych et al., 2024; Adila et al., 2024; Lu et al., 2024; An et al., 2024). *Explicit debiasing* refers to debiasing concepts chosen by the user or predefined policies. *Unsupervised* indicates access only to unlabeled images at test time. *Episodic* denotes processing a single image at a time, as opposed to batch processing in an online setting. Given the absence of differences from our setup, ORTHCALI serves as our fairness baseline.

The zero-shot classification procedure is summarized in Algorithm 1.

---
**Algorithm 1** Zero-Shot classification with VLMs

---
1: **Input:** test image $x$, and vision-language model $\{f_V, f_L\}$.
2: construct the soft-prompt set $\{t_{ctx}; \mathcal{Y}\} = \{(t_{ctx}, t_y) : y \in \mathcal{Y}\}$
3: obtain the set of text features $\{l_y = f_L((t_{ctx}, t_y)) : y \in \mathcal{Y}\}$ using the text encoder
4: obtain the image feature $v = f_V(x)$ using the image encoder
5: compute the similarity score $\texttt{sim}(l_y, v)$ for each pair $(l_y, v)$
6: obtain the softmax probability distribution $p(\cdot \mid x, \{t_{ctx}; \mathcal{Y}\}, \tau)$
7: sample $\widehat{y} \sim p(\cdot | x, \{t_{ctx}; \mathcal{Y}\}, \tau)$
8: **Output:** predicted target label $\widehat{y}$.

---

The test-time adaptive classification procedure is summarized in Algorithm 2.

---
**Algorithm 2** Test-Time Adaptive classification with VLMs

---
1: **Input:** test image $x$, vision-language model $\{f_V, f_L\}$, and set of augmentation functions $\mathcal{A} = \{A_1, \ldots, A_N\}$.
2: construct the soft-prompt set $\{t_{ctx}; \mathcal{Y}\} = \{(t_{ctx}, t_y) : y \in \mathcal{Y}\}$
3: ▷ TPT
4: **Require:** update rule $G$ with learning rate $\eta$, and number of epochs $n_{epochs}$
5: compute the marginal probability distribution $\overline{p}(\cdot \mid x, \{t_{ctx}; \mathcal{Y}\}, \tau)$ using $\mathcal{A}_{filtered}$
6: if ELRA: set the learning rate to $\eta_{ELRA}$ using Algorithm 4
7: update the shared soft-prompt by optimizing Equation 1 via $G$ for $n_{epochs}$ to obtain $t^*_{ctx}$
8: obtain $\widehat{y}$ by using Algorithm 1 with the updated set $\{t^*_{ctx}; \mathcal{Y}\}$
9: ▷ ZERO
10: obtain $\widehat{y} = \text{ZERO}(x, \{t_{ctx}; \mathcal{Y}\})$ using $\mathcal{A}_{filtered}$ (see Equation 2)
11: **Output:** predicted target label $\widehat{y}$.

---

The fairness-aware test-time adaptive classification procedure is summarized in Algorithm 3.

The entropic learning rate adapter procedure is summarized in Algorithm 4.

The hyperparameters used in all methods are summarized in Table 3.

## A.2 DATASETS

Tables 4 and 5 show how initial prompts are constructed for the different datasets.

---

**Algorithm 3** Fairness-Aware Test-Time Adaptive classification with VLMs

---

1: **Input:** test image $x$, vision-language model $\{f_V, f_L\}$, and set of augmentation functions $\mathcal{A} = \{A_1, \ldots, A_N\}$.
2: construct the target soft-prompt set $\{\boldsymbol{t}_{\text{ctx}}; \mathcal{Y}\} = \{(\boldsymbol{t}_{\text{ctx}}, \boldsymbol{t}_y) : y \in \mathcal{Y}\}$
3: construct the spurious soft-prompt set $\{\boldsymbol{t}_{\text{ctx}}; \mathcal{S}\} = \{(\boldsymbol{t}_{\text{ctx}}, \boldsymbol{t}_s) : s \in \mathcal{S}\}$
4: ▷ ORTHCALI
5: **Require:** orthogonal calibration strength $\lambda_{\text{orth}}$
6: define a matrix $A$ whose columns are the embeddings of spurious prompts, and the orthogonal projection matrix $P_0 = I - A(A^\top A)^{-1} A^\top$
7: construct the target-spurious joint soft-prompt set $\{\boldsymbol{t}_{\text{ctx}}, \boldsymbol{t}_y; \mathcal{S}\}$ for each target $y$
8: construct a set $\mathcal{D}$ containing all positive pairs of embeddings (i.e., pairs with the same target but different spurious attributes)
9: optimize the projection matrix using $\mathcal{D}$:

$$P^* \;\leftarrow\; \arg\min_P \|P - P_0\|^2 + \frac{\lambda_{\text{orth}}}{|\mathcal{D}|} \sum_{(\boldsymbol{z}_i, \boldsymbol{z}_j) \in \mathcal{D}} \|P\boldsymbol{z}_i - P\boldsymbol{z}_j\|^2 ,$$

10: project target soft-prompts $\{\boldsymbol{t}_{\text{ctx}}; \mathcal{Y}\}$ using $P^*$
11: obtain $\widehat{y}$ by using Algorithm 1 with the projected target soft-prompts
12: ▷ FAIRTPT
13: **Require:** update rule $G$ with learning rate $\eta$, and number of epochs $n_{\text{epochs}}$
14: compute the target marginal probability distribution $\overline{p}(\cdot \mid x, \{\boldsymbol{t}_{\text{ctx}}; \mathcal{Y}\}, \tau)$ using $\mathcal{A}_{\text{filtered}}$
15: compute the spurious marginal probability distribution $\overline{p}(\cdot \mid x, \{\boldsymbol{t}_{\text{ctx}}; \mathcal{S}\}, \tau)$ using $\mathcal{A}_{\text{filtered}}$
16: if ELRA: set the learning rate to $\eta_{\text{ELRA}}$ using Algorithm 4
17: update the shared soft-prompt by optimizing Equation 3 (or Equation 6 for FAIRTPT (MO)) via $G$ for $n_{\text{epochs}}$ to obtain $\boldsymbol{t}_{\text{ctx}}^*$
18: use Algorithm 1 with the updated set $\{\boldsymbol{t}_{\text{ctx}}^*; \mathcal{Y}\}$ to obtain $\widehat{y}$
19: **Output:** predicted target label $\widehat{y}$.

---

**Algorithm 4** Entropic Learning Rate Adapter

---

1: **Input:** initial context $\boldsymbol{t}_{\text{ctx}}$, desired target loss change amplitude $\beta$, auto differentiation method $B$, and target entropic loss $\ell_{\text{ent}} : \boldsymbol{t}_{\text{ctx}} \to \mathbb{R}$
2: set optimizer $G$ as SGD
3: call $B$ to obtain the initial gradient $\boldsymbol{g}_{\text{init}}$ of $\boldsymbol{t}_{\text{ctx}}$
4: set the learning rate to $\eta = 0.01 \cdot \|\boldsymbol{t}_{\text{ctx}}\| / \|\boldsymbol{g}_{\text{init}}\|$ to keep $G$ in the linear regime
5: reset gradients
6: call $B$ and perform an optimizer step to obtain the new context $\boldsymbol{t}_{\text{ctx}}'$
7: compute $\Delta\ell_{\text{ent}} = |\ell_{\text{ent}}(\boldsymbol{t}_{\text{ctx}}) - \ell_{\text{ent}}(\boldsymbol{t}_{\text{ctx}}')|$
8: reset the optimizer state
9: **Output:** adapted learning rate $\eta_{\text{ELRA}} = \eta \times \beta / \Delta\ell_{\text{ent}}$

---

## A.3 RESULTS

Tables 6, 7, and 8 present the $(\eta, \rho)$ benchmark of TPT, the $\rho$ benchmark of ZERO, and the $\lambda_{\text{orth}}$ benchmark of ORTHCALI, respectively, w.r.t. overall and subgroup-level performance metrics.

Tables 9 and 10 benchmark FAIRTPT for diverse values of $\lambda_{\text{fair}}$ by ablating ELRA and changing the loss respectively.

Table 11 presents overall and subgroup-level performance evaluation of FAIRTPT followed by the orthogonal projection of ORTHCALI.

Table 12 presents the benchmark of all methods on additional dataset-attribute configurations.

| Method | $A$ | $\rho$ | Optimizer | $\eta_{\text{init}}$ | $\eta$ | $\beta$ | $n_{\text{epochs}}$ | $\lambda$ |
|---|---|---|---|---|---|---|---|---|
| TPT | 64 | 0.1 | AdamW | | $5e-3$ | | 1 | |
| ZERO | 64 | 0.1 | | | | | | |
| ORTHCALI | | | | | | | | $\lambda_{\text{orth}} = 1000$ |
| FAIRTPT | 64 | 0.75 | SGD | Any | $\eta_{\text{ELRA}}$ | 0.01 | 1 | $\lambda_{\text{fair}} = 100$ |
| FAIRTPT (MO) | 64 | 0.75 | UPGrad | Any | $\eta_{\text{ELRA}}$ | 0.01 | 1 | $\lambda_{\text{fair (mo)}} = 100$ |
| FAIRTPT w/o ELRA | 64 | 0.75 | AdamW | | $5e-3$ | | 1 | $\lambda_{\text{fair}} = 100$ |
| FAIRTPT (MO) w/o ELRA | 64 | 0.75 | UPGrad | | $5e-3$ | | 1 | $\lambda_{\text{fair (mo)}} = 100$ |

Table 3: Hyperparameters of all the methods evaluated in our experiments.

| Dataset | Loss Term | Initial Prompt Template |
|---|---|---|
| FairFace (Karkkainen & Joo, 2021) | $\ell_{\text{ent}}(x, \{t_{\text{ctx}}; \mathcal{Y}\})$ 
 $\ell_{\text{ent}}(x, \{t_{\text{ctx}}; \mathcal{S}\})$ 
 $\ell_{\text{ent}}(x, \{t_{\text{ctx}}, t_y; \mathcal{S}\})$ | `A photo of a [T-CLS] person.` 
 `A photo of a [S-CLS] person.` 
 `A photo of a [S-CLS] [T-CLS] person.` |
| CelebA (Liu et al., 2015) | $\ell_{\text{ent}}(x, \{t_{\text{ctx}}; \mathcal{Y}\})$ 
 $\ell_{\text{ent}}(x, \{t_{\text{ctx}}; \mathcal{S}\})$ 
 $\ell_{\text{ent}}(x, \{t_{\text{ctx}}, t_y; \mathcal{S}\})$ | `A photo of a celebrity [T-CLS].` 
 `A photo of a [S-CLS] celebrity.` 
 `A photo of a [S-CLS] celebrity [T-CLS].` |
| WaterBirds (Sagawa et al., 2020) | $\ell_{\text{ent}}(x, \{t_{\text{ctx}}; \mathcal{Y}\})$ 
 $\ell_{\text{ent}}(x, \{t_{\text{ctx}}; \mathcal{S}\})$ 
 $\ell_{\text{ent}}(x, \{t_{\text{ctx}}, t_y; \mathcal{S}\})$ | `This is a picture [T-CLS].` 
 `This is a picture [S-CLS].` 
 `This is a picture [T-CLS] [S-CLS].` |
| UTKFace (Zhang et al., 2017) | $\ell_{\text{ent}}(x, \{t_{\text{ctx}}; \mathcal{Y}\})$ 
 $\ell_{\text{ent}}(x, \{t_{\text{ctx}}; \mathcal{S}\})$ 
 $\ell_{\text{ent}}(x, \{t_{\text{ctx}}, t_y; \mathcal{S}\})$ | `A photo of a [T-CLS] person.` 
 `A photo of a [S-CLS] person.` 
 `A photo of a [S-CLS] [T-CLS] person.` |

Table 4: Initial prompt templates for each dataset. The placeholders `[T-CLS]` and `[S-CLS]` denote the target and spurious attributes, respectively, as defined in Table 5.

| Dataset | Target Attribute (`[T-CLS]` String) | Spurious Attribute (`[S-CLS]` String) |
|---|---|---|
| FairFace | Gender (`male`, `female`) | Race (`White`, `Southeast Asian`, `Middle Eastern`, `Black`, `Indian`, `Latino Hispanic`, `East Asian`) |
| CelebA | Hair color (`with dark hair`, `with blond hair`) 
 Smile (`smiling`, `not smiling`) | Gender (`male`, `female`) |
| WaterBirds | Bird type (`of a water bird`, `of a land bird`) | Background (`on water`, `on land`) |
| UTKFace | Age (`young`, `old`) 
 Gender (`male`, `female`) | Race (`White`, `Black`, `Asian`, `Indian`) |

Table 5: Placeholder values for the prompt templates shown in Table 4.

| METHOD | FAIRFACE | | | | CELEBA | | | | | | | | WATERBIRDS | | | | UTKFACE | | | | | | | | AVERAGE RESULTS | | | |
|---|---|---|---|---|---|---|---|---|---|---|---|---|---|---|---|---|---|---|---|---|---|---|---|---|---|---|---|---|
| | Gender × Race | | | | Hair color × Gender | | | | Smiling × Gender | | | | Type × Background | | | | Age × Race | | | | Gender × Race | | | | | | | |
| | A | WGA | B | EOD | A | WGA | B | EOD | A | WGA | B | EOD | A | WGA | B | EOD | A | WGA | B | EOD | A | WGA | B | EOD | A | WGA | B | EOD |
| Zero Shot | 95.7 | 90.0 | 5.7 | 9.4 | 86.4 | 67.8 | 18.6 | 22.3 | 75.8 | 53.7 | 22.1 | 8.0 | 83.8 | 40.2 | 43.7 | 25.0 | 80.3 | 45.7 | 34.5 | 23.2 | 97.1 | 90.1 | 7.0 | 9.1 | 86.5 | 64.6 | 21.9 | 16.2 |
| **η = 1e − 5** | | | | | | | | | | | | | | | | | | | | | | | | | | | | |
| ρ = 0.1 | 95.7 | 90.5 | 5.2 | 8.9 | 86.5 | 67.8 | 18.8 | 22.3 | 75.5 | 52.9 | 22.6 | 8.3 | 83.8 | 40.2 | 43.7 | 25.0 | 80.3 | 45.6 | 34.6 | 23.3 | 97.1 | 90.3 | 6.8 | 8.9 | 86.5 | 64.5 | 21.9 | 16.1 |
| ρ = 0.25 | 95.7 | 90.3 | 5.4 | 9.1 | 86.5 | 67.8 | 18.8 | 22.3 | 75.5 | 52.9 | 22.6 | 8.3 | 83.9 | 40.2 | 43.7 | 25.2 | 80.3 | 45.6 | 34.6 | 23.3 | 97.1 | 90.3 | 6.8 | 8.9 | 86.5 | 64.5 | 22.0 | 16.2 |
| ρ = 0.5 | 95.7 | 90.3 | 5.4 | 9.1 | 86.6 | 67.8 | 18.8 | 22.3 | 75.5 | 52.8 | 22.7 | 8.4 | 83.8 | 40.2 | 43.7 | 25.0 | 80.3 | 45.6 | 34.6 | 23.3 | 97.1 | 90.3 | 6.8 | 8.9 | 86.5 | 64.5 | 22.0 | 16.2 |
| ρ = 0.75 | 95.7 | 90.3 | 5.4 | 9.1 | 86.6 | 67.8 | 18.8 | 22.6 | 75.5 | 52.9 | 22.7 | 8.3 | 83.9 | 40.2 | 43.7 | 25.0 | 80.3 | 45.7 | 34.5 | 23.2 | 97.1 | 90.3 | 6.8 | 8.9 | 86.5 | 64.5 | 22.0 | 16.2 |
| ρ = 1.0 | 95.7 | 90.3 | 5.4 | 9.1 | 86.6 | 67.8 | 18.8 | 22.5 | 75.5 | 52.8 | 22.8 | 8.4 | 83.9 | 40.2 | 43.7 | 25.2 | 80.3 | 45.6 | 34.6 | 23.3 | 97.1 | 90.3 | 6.8 | 8.9 | 86.5 | 64.5 | 22.0 | 16.2 |
| **η = 1e − 4** | | | | | | | | | | | | | | | | | | | | | | | | | | | | |
| ρ = 0.1 | 95.7 | 90.5 | 5.1 | 8.9 | 87.4 | 61.0 | 26.4 | 28.0 | 73.6 | 48.2 | 25.4 | 8.7 | 83.9 | 39.3 | 44.6 | 26.6 | 81.1 | 46.8 | 34.3 | 24.6 | 97.1 | 91.1 | 6.0 | 8.1 | 86.5 | 62.8 | 23.6 | 17.5 |
| ρ = 0.25 | 95.7 | 90.4 | 5.3 | 9.0 | 87.7 | 61.1 | 26.5 | 28.2 | 73.7 | 48.3 | 25.4 | 8.7 | 83.9 | 39.3 | 44.6 | 26.6 | 81.5 | 47.2 | 34.3 | 24.9 | 97.1 | 91.1 | 6.0 | 8.1 | 86.6 | 62.9 | 23.7 | 17.6 |
| ρ = 0.5 | 95.7 | 90.4 | 5.3 | 9.0 | 87.6 | 61.2 | 26.4 | 28.1 | 73.7 | 47.8 | 25.9 | 9.6 | 83.9 | 39.5 | 44.4 | 26.1 | 81.6 | 47.4 | 34.2 | 24.7 | 97.1 | 91.1 | 6.0 | 8.2 | 86.6 | 62.9 | 23.7 | 17.6 |
| ρ = 0.75 | 95.7 | 90.4 | 5.3 | 9.0 | 87.6 | 61.2 | 26.4 | 28.1 | 73.6 | 47.8 | 25.8 | 9.4 | 83.8 | 39.6 | 44.2 | 25.7 | 81.7 | 47.5 | 34.2 | 24.6 | 97.1 | 91.1 | 6.0 | 8.2 | 86.6 | 62.9 | 23.6 | 17.5 |
| ρ = 1.0 | 95.7 | 90.3 | 5.4 | 9.3 | 87.6 | 61.2 | 26.4 | 27.9 | 73.6 | 47.5 | 26.0 | 9.7 | 83.8 | 39.8 | 44.0 | 25.6 | 81.8 | 47.6 | 34.2 | 24.5 | 97.1 | 91.1 | 6.0 | 8.2 | 86.6 | 62.9 | 23.7 | 17.5 |
| **η = 1e − 3** | | | | | | | | | | | | | | | | | | | | | | | | | | | | |
| ρ = 0.1 | 95.1 | 88.4 | 6.7 | 9.8 | 91.8 | 42.6 | 49.2 | 38.7 | 59.2 | 17.4 | 41.8 | 9.8 | 84.2 | 44.2 | 40.0 | 33.2 | 86.1 | 57.4 | 28.7 | 23.4 | 95.9 | 89.6 | 6.3 | 7.7 | 85.4 | 56.6 | 28.8 | 20.4 |
| ρ = 0.25 | 94.7 | 88.1 | 6.6 | 9.4 | 91.3 | 56.0 | 35.2 | 29.3 | 62.7 | 20.6 | 42.0 | 16.2 | 83.6 | 42.6 | 41.1 | 33.3 | 86.6 | 55.8 | 30.8 | 26.3 | 96.2 | 89.8 | 6.3 | 7.1 | 85.9 | 58.8 | 27.0 | 20.3 |
| ρ = 0.5 | 95.1 | 88.6 | 6.5 | 9.9 | 90.7 | 56.3 | 34.4 | 27.8 | 65.3 | 26.4 | 38.9 | 15.2 | 83.8 | 42.1 | 41.8 | 33.6 | 86.7 | 55.4 | 31.3 | 31.4 | 96.1 | 89.1 | 7.1 | 8.2 | 86.3 | 59.6 | 26.7 | 21.0 |
| ρ = 0.75 | 94.9 | 87.5 | 7.5 | 10.4 | 90.5 | 56.3 | 34.2 | 27.7 | 65.9 | 27.7 | 38.2 | 14.9 | 83.9 | 42.1 | 41.8 | 33.3 | 86.7 | 55.8 | 30.9 | 30.5 | 96.1 | 89.1 | 7.1 | 8.3 | 86.4 | 59.7 | 26.6 | 20.8 |
| ρ = 1.0 | 94.9 | 87.5 | 7.5 | 10.5 | 90.4 | 56.3 | 34.1 | 27.4 | 66.0 | 27.7 | 38.2 | 14.9 | 84.2 | 42.6 | 41.6 | 33.0 | 86.8 | 55.4 | 31.4 | 31.4 | 96.2 | 89.2 | 7.1 | 8.2 | 86.4 | 59.8 | 26.6 | 20.9 |
| **η = 1e − 2** | | | | | | | | | | | | | | | | | | | | | | | | | | | | |
| ρ = 0.1 | 93.5 | 82.9 | 10.6 | 15.7 | 91.8 | 42.6 | 49.2 | 38.1 | 60.0 | 20.4 | 39.6 | 6.9 | 83.2 | 41.7 | 41.5 | 36.5 | 85.3 | 59.3 | 26.0 | 26.8 | 94.6 | 87.1 | 7.5 | 11.1 | 84.8 | 55.7 | 29.1 | 22.5 |
| ρ = 0.25 | 94.3 | 86.7 | 7.6 | 12.6 | 91.2 | 56.0 | 35.1 | 29.4 | 63.5 | 23.6 | 39.9 | 13.4 | 82.9 | 40.9 | 42.0 | 36.5 | 86.1 | 57.2 | 28.9 | 28.4 | 95.7 | 89.0 | 6.6 | 8.5 | 85.6 | 58.9 | 26.7 | 21.5 |
| ρ = 0.5 | 94.3 | 87.1 | 7.2 | 11.4 | 90.6 | 56.3 | 34.3 | 27.8 | 66.2 | 29.1 | 37.1 | 13.0 | 83.4 | 41.6 | 41.8 | 34.0 | 85.9 | 56.2 | 29.7 | 30.8 | 95.7 | 89.0 | 6.7 | 9.0 | 86.0 | 59.9 | 26.1 | 21.0 |
| ρ = 0.75 | 94.3 | 87.0 | 7.3 | 11.9 | 90.4 | 56.3 | 34.1 | 27.7 | 66.5 | 29.7 | 36.8 | 13.1 | 83.6 | 41.7 | 41.9 | 32.8 | 85.9 | 58.0 | 28.0 | 29.3 | 95.7 | 87.8 | 7.9 | 10.3 | 86.1 | 60.1 | 26.0 | 20.8 |
| ρ = 1.0 | 94.3 | 87.8 | 6.5 | 10.5 | 90.3 | 56.3 | 34.0 | 27.4 | 66.4 | 29.2 | 37.2 | 13.7 | 83.9 | 42.2 | 41.6 | 32.6 | 85.9 | 57.1 | 28.8 | 29.1 | 95.5 | 88.1 | 7.3 | 10.2 | 86.0 | 60.1 | 25.9 | 20.6 |
| **η = 1e − 1** | | | | | | | | | | | | | | | | | | | | | | | | | | | | |
| ρ = 0.1 | 82.2 | 70.6 | 11.6 | 19.2 | 65.0 | 55.5 | 9.5 | 22.2 | 64.0 | 44.3 | 19.7 | 12.4 | 75.6 | 26.8 | 48.8 | 25.7 | 70.5 | 52.8 | 17.7 | 26.6 | 81.3 | 66.0 | 15.3 | 16.1 | 73.1 | 52.7 | 20.4 | 20.3 |
| ρ = 0.25 | 82.1 | 73.5 | 8.6 | 16.3 | 67.3 | 58.5 | 8.8 | 12.8 | 64.8 | 40.8 | 24.0 | 14.9 | 76.8 | 27.5 | 49.3 | 29.6 | 69.6 | 51.7 | 18.0 | 26.7 | 81.1 | 61.7 | 19.4 | 18.0 | 73.6 | 52.3 | 21.4 | 19.7 |
| ρ = 0.5 | 82.8 | 72.1 | 10.7 | 17.6 | 66.8 | 56.7 | 10.1 | 20.9 | 65.3 | 39.6 | 25.8 | 17.5 | 77.1 | 26.1 | 51.0 | 32.6 | 70.7 | 52.7 | 16.9 | 25.6 | 80.7 | 61.7 | 19.0 | 18.1 | 73.7 | 51.5 | 22.2 | 22.0 |
| ρ = 0.75 | 83.0 | 74.1 | 8.9 | 14.5 | 68.3 | 59.2 | 9.0 | 20.6 | 65.3 | 38.3 | 27.0 | 17.9 | 77.0 | 26.1 | 50.8 | 34.0 | 70.4 | 52.2 | 18.2 | 30.0 | 81.1 | 63.2 | 17.9 | 16.4 | 74.2 | 52.2 | 22.0 | 22.2 |
| ρ = 1.0 | 80.7 | 71.9 | 8.8 | 16.5 | 68.5 | 60.6 | 7.9 | 19.2 | 65.5 | 37.7 | 27.7 | 18.7 | 77.0 | 29.7 | 47.3 | 37.4 | 70.9 | 57.4 | 13.5 | 21.6 | 79.8 | 58.6 | 21.2 | 20.4 | 73.7 | 52.6 | 21.1 | 22.3 |

Table 6: Hyperparameter benchmark of TPT. We report mean over 5 random seeds and mean aggregation over all equally sized datasets. Best results are in **bold**, second best underlined. The values that improve upon ZERO-SHOT by more than 2.0 percentage points are highlighted in green, while degradations greater than 2.0 percentage points are shown in red.

| METHOD | FAIRFACE | | | | CELEBA | | | | | | | | WATERBIRDS | | | | UTKFACE | | | | | | | | AVERAGE RESULTS | | | |
|---|---|---|---|---|---|---|---|---|---|---|---|---|---|---|---|---|---|---|---|---|---|---|---|---|---|---|---|---|
| | Gender × Race | | | | Hair color × Gender | | | | Smiling × Gender | | | | Type × Background | | | | Age × Race | | | | Gender × Race | | | | | | | |
| | A | WGA | B | EOD | A | WGA | B | EOD | A | WGA | B | EOD | A | WGA | B | EOD | A | WGA | B | EOD | A | WGA | B | EOD | A | WGA | B | EOD |
| Zero Shot | 95.7 | 90.0 | 5.7 | 9.4 | 86.4 | 67.8 | 18.6 | 22.3 | 75.8 | 53.7 | 22.1 | 8.0 | 83.8 | 40.2 | 43.7 | 25.0 | 80.3 | 45.7 | 34.5 | 23.2 | 97.1 | 90.1 | 7.0 | 9.1 | 86.5 | 64.6 | 21.9 | 16.2 |
| ρ = 0.1 | 91.2 | 78.6 | 12.6 | 11.9 | 90.4 | 56.3 | 34.0 | 23.8 | 69.9 | 38.0 | 31.9 | 11.1 | 83.1 | 40.0 | 43.2 | 29.0 | 86.2 | 53.8 | 32.4 | 28.7 | 93.5 | 77.8 | 15.7 | 13.4 | 85.7 | 57.4 | 28.3 | 19.6 |
| ρ = 0.25 | 93.4 | 83.3 | 10.1 | 11.4 | 90.4 | 56.3 | 34.1 | 27.1 | 68.7 | 34.3 | 34.4 | 12.6 | 83.9 | 41.9 | 42.0 | 31.0 | 86.6 | 56.0 | 30.5 | 30.3 | 94.8 | 81.4 | 13.4 | 12.9 | 86.3 | 58.9 | 27.4 | 20.9 |
| ρ = 0.5 | 94.1 | 86.2 | 7.8 | 9.3 | 90.3 | 56.3 | 34.0 | 28.0 | 68.2 | 33.1 | 35.1 | 13.2 | 84.1 | 42.1 | 42.0 | 31.1 | 87.0 | 58.3 | 28.7 | 28.1 | 94.9 | 80.2 | 14.7 | 14.2 | 86.4 | 59.4 | 27.1 | 20.6 |
| ρ = 0.75 | 94.2 | 86.1 | 8.1 | 9.8 | 90.2 | 56.3 | 33.9 | 27.7 | 68.2 | 32.5 | 35.7 | 13.7 | 84.1 | 43.1 | 41.0 | 29.0 | 87.0 | 57.0 | 30.0 | 29.0 | 95.1 | 80.5 | 14.6 | 14.2 | 86.5 | 59.2 | 27.2 | 20.6 |
| ρ = 1.0 | 94.3 | 87.0 | 7.3 | 8.9 | 90.2 | 56.3 | 33.8 | 27.8 | 68.5 | 32.9 | 35.6 | 14.2 | 84.2 | 42.6 | 41.6 | 30.2 | 87.1 | 56.4 | 30.7 | 31.1 | 95.1 | 79.9 | 15.2 | 15.0 | 86.5 | 59.2 | 27.4 | 21.2 |

Table 7: Hyperparameter benchmark of ZERO. We report mean over 5 random seeds and mean aggregation over all equally sized datasets. Best results are in **bold**, second best underlined. The values that improve upon ZERO-SHOT by more than 2.0 percentage points are highlighted in green, while degradations greater than 2.0 percentage points are shown in red.

| METHOD | FAIRFACE | | | | CELEBA | | | | | | | | WATERBIRDS | | | | UTKFACE | | | | | | | | AVERAGE RESULTS | | | |
|---|---|---|---|---|---|---|---|---|---|---|---|---|---|---|---|---|---|---|---|---|---|---|---|---|---|---|---|---|
| | Gender × Race | | | | Hair color × Gender | | | | Smiling × Gender | | | | Type × Background | | | | Age × Race | | | | Gender × Race | | | | | | | |
| | A | WGA | B | EOD | A | WGA | B | EOD | A | WGA | B | EOD | A | WGA | B | EOD | A | WGA | B | EOD | A | WGA | B | EOD | A | WGA | B | EOD |
| Zero Shot | 95.7 | 90.0 | 5.7 | 9.4 | 86.4 | 67.8 | 18.6 | 22.3 | 75.8 | 53.7 | 22.1 | 8.0 | 83.8 | 40.2 | 43.7 | 25.0 | 80.3 | 45.7 | 34.5 | 23.2 | 97.1 | 90.1 | 7.0 | 9.1 | 86.5 | 64.6 | 21.9 | 16.2 |
| $\lambda_{\text{orth}} = 0$ | 96.2 | 90.9 | 5.3 | 8.5 | 86.2 | 70.7 | 15.5 | 22.7 | 71.8 | 40.1 | 31.7 | 24.4 | 85.5 | 55.7 | 29.9 | 4.3 | 81.1 | 45.2 | 36.0 | 18.9 | 96.9 | 89.6 | 7.3 | 9.4 | 86.3 | 65.4 | 20.9 | 14.7 |
| $\lambda_{\text{orth}} = 1$ | 96.2 | 90.9 | 5.3 | 8.5 | 86.2 | 70.7 | 15.5 | 22.7 | 71.7 | 39.9 | 31.8 | 24.5 | 86.0 | 58.5 | 27.5 | 3.6 | 81.1 | 45.2 | 36.0 | 18.2 | 96.9 | 89.6 | 7.3 | 9.4 | 86.4 | 65.8 | 20.6 | 14.5 |
| $\lambda_{\text{orth}} = 10$ | 96.1 | 90.6 | 5.5 | 8.8 | 86.1 | 70.7 | 15.4 | 22.5 | 71.5 | 38.3 | 33.2 | 27.6 | 86.5 | 63.1 | 23.4 | 8.5 | 81.1 | 45.2 | 36.0 | 18.6 | 97.0 | 89.6 | 7.3 | 9.4 | 86.4 | 66.3 | 20.1 | 15.9 |
| $\lambda_{\text{orth}} = 1000$ | 95.9 | 87.8 | 8.1 | 11.3 | 85.1 | 71.6 | 13.5 | 22.0 | 71.4 | 36.8 | 34.6 | 30.6 | 83.3 | 62.0 | 21.4 | 16.2 | 79.8 | 47.0 | 32.8 | 14.4 | 96.8 | 88.9 | 7.9 | 10.1 | 85.4 | 65.7 | 19.7 | 17.4 |
| $\lambda_{\text{orth}} = 100000$ | 95.8 | 86.5 | 9.3 | 12.7 | 85.0 | 71.6 | 13.4 | 22.0 | 71.4 | 36.4 | 35.0 | 31.2 | 83.0 | 56.4 | 26.7 | 13.9 | 79.4 | 47.9 | 31.5 | 13.9 | 96.8 | 88.7 | 8.2 | 10.3 | 85.2 | 64.6 | 20.7 | 17.3 |

Table 8: Hyperparameter benchmark of ORTHCALI. We report mean over 5 random seeds and mean aggregation over all equally sized datasets. Best results are in **bold**, second best underlined. The values that improve upon ZERO-SHOT by more than 2.0 percentage points are highlighted in green, while degradations greater than 2.0 percentage points are shown in red.

| METHOD | FAIRFACE Gender × Race | | | | CELEBA Hair color × Gender | | | | Smiling × Gender | | | | WATERBIRDS Type × Background | | | | UTKFACE Age × Race | | | | Gender × Race | | | | AVERAGE RESULTS | | | |
|---|---|---|---|---|---|---|---|---|---|---|---|---|---|---|---|---|---|---|---|---|---|---|---|---|---|---|---|---|---|
| | A | WGA | B | EOD | A | WGA | B | EOD | A | WGA | B | EOD | A | WGA | B | EOD | A | WGA | B | EOD | A | WGA | B | EOD | A | WGA | B | EOD |
| Zero Shot | 95.7 | 90.0 | 5.7 | 9.4 | 86.4 | 67.7 | 18.6 | 22.3 | 75.8 | 53.7 | 22.1 | 8.0 | 83.8 | 40.2 | 43.7 | 25.0 | 80.3 | 45.7 | 34.5 | 23.2 | 97.1 | 90.1 | 7.0 | 9.1 | 86.5 | 64.6 | 21.9 | 16.2 |
| **FAIRTPT** | | | | | | | | | | | | | | | | | | | | | | | | | | | | |
| $\lambda_{\text{fair}}=1$ | 95.5 | 90.5 | 5.0 | 8.8 | 86.2 | 67.7 | 18.5 | 22.0 | 75.3 | 52.6 | 22.7 | 8.1 | 83.9 | 41.0 | 42.8 | 26.2 | 80.3 | 46.5 | 33.8 | 22.5 | 97.0 | 90.6 | 6.5 | 8.7 | 86.4 | 64.8 | 21.6 | 16.1 |
| $\lambda_{\text{fair}}=100$ | 95.5 | 90.7 | 4.8 | 8.1 | 85.6 | 67.7 | 17.9 | 21.9 | 75.9 | 56.1 | 19.8 | 6.2 | 83.2 | 42.4 | 40.8 | 23.5 | 81.0 | 51.5 | 29.5 | 23.4 | 96.7 | 90.6 | 6.1 | 8.6 | 86.3 | 66.5 | 19.8 | 15.3 |
| $\lambda_{\text{fair}}=5000$ | 95.4 | 90.6 | 4.9 | 8.1 | 85.5 | 67.5 | 18.0 | 21.5 | 75.9 | 55.6 | 20.3 | 5.8 | 83.3 | 43.6 | 39.7 | 23.4 | 80.2 | 49.4 | 30.8 | 26.4 | 96.7 | 90.1 | 6.6 | 9.0 | 86.2 | 66.1 | 20.0 | 15.7 |
| **FAIRTPT (MO)** | | | | | | | | | | | | | | | | | | | | | | | | | | | | |
| $\lambda_{\text{fair (mo)}}=1$ | 95.4 | 90.0 | 5.4 | 9.5 | 86.3 | 67.8 | 18.6 | 22.1 | 75.3 | 52.3 | 23.0 | 8.4 | 83.8 | 40.9 | 42.9 | 26.2 | 80.3 | 46.7 | 33.6 | 22.3 | 97.0 | 90.6 | 6.5 | 8.7 | 86.4 | 64.7 | 21.7 | 16.2 |
| $\lambda_{\text{fair (mo)}}=100$ | 95.3 | 90.4 | 4.9 | 7.9 | 85.3 | 67.5 | 17.8 | 21.3 | 76.1 | 57.8 | 18.3 | 6.3 | 83.2 | 41.4 | 41.8 | 24.6 | 80.8 | 51.6 | 29.3 | 20.9 | 96.6 | 90.9 | 5.8 | 8.1 | 86.2 | 66.6 | 19.6 | 14.8 |
| $\lambda_{\text{fair (mo)}}=5000$ | 95.3 | 90.2 | 5.2 | 8.4 | 85.2 | 67.4 | 17.8 | 22.0 | 76.0 | 56.3 | 19.7 | 6.2 | 83.2 | 41.7 | 41.5 | 24.8 | 80.9 | 48.9 | 32.0 | 26.3 | 96.7 | 90.3 | 6.3 | 8.5 | 86.2 | 65.8 | 20.4 | 16.0 |
| **FAIRTPT** *special cases* | | | | | | | | | | | | | | | | | | | | | | | | | | | | |
| $\lambda_{\text{fair}}=0$ | 95.1 | 90.1 | 5.0 | 9.2 | 86.9 | 65.8 | 21.1 | 23.9 | 74.4 | 50.5 | 23.9 | 8.1 | 83.7 | 40.2 | 43.5 | 25.2 | 80.3 | 46.2 | 34.1 | 22.8 | 96.6 | 90.3 | 6.3 | 8.8 | 86.2 | 63.8 | 22.3 | 16.4 |
| $\lambda_{\text{fair}}=\infty$ | 95.5 | 90.1 | 5.4 | 9.0 | 85.2 | 67.4 | 17.8 | 21.6 | 75.9 | 55.8 | 20.1 | 5.9 | 83.5 | 44.5 | 39.0 | 25.0 | 80.8 | 49.7 | 31.1 | 27.6 | 96.8 | 90.3 | 6.5 | 8.9 | 86.3 | 66.3 | 20.0 | 15.8 |
| **FAIRTPT** *without ELRA* | | | | | | | | | | | | | | | | | | | | | | | | | | | | |
| $\lambda_{\text{fair}}=1$ | 94.5 | 79.6 | 14.9 | 17.6 | 89.9 | 56.3 | 33.6 | 28.8 | 67.7 | 30.2 | 37.5 | 16.8 | 84.4 | 43.6 | 40.8 | 30.3 | 87.0 | 56.4 | 30.6 | 31.8 | 95.7 | 88.4 | 7.3 | 8.5 | 86.5 | 59.1 | 27.4 | 22.3 |
| $\lambda_{\text{fair}}=100$ | 88.6 | 25.2 | 63.3 | 71.9 | 68.7 | 51.1 | 17.6 | 25.3 | 79.0 | 65.9 | 13.1 | 25.0 | 84.3 | 45.2 | 39.1 | 23.3 | 76.5 | 62.4 | 14.2 | 30.9 | 82.4 | 15.2 | 67.2 | 80.9 | 79.9 | 44.2 | 35.8 | 39.4 |
| $\lambda_{\text{fair}}=5000$ | 88.2 | 24.8 | 63.4 | 72.4 | 67.3 | 50.3 | 17.0 | 23.5 | 80.4 | 25.0 | 25.0 | 6.1 | 84.1 | 45.4 | 38.7 | 22.7 | 75.2 | 62.3 | 25.0 | 29.2 | 82.0 | 13.7 | 68.2 | 82.0 | 79.5 | 44.3 | 35.2 | 39.3 |
| **FAIRTPT (MO)** *without ELRA* | | | | | | | | | | | | | | | | | | | | | | | | | | | | |
| $\lambda_{\text{fair (mo)}}=1$ | 94.5 | 80.4 | 14.2 | 16.5 | 90.0 | 56.3 | 33.7 | 28.8 | 67.6 | 29.7 | 37.9 | 17.4 | 84.4 | 43.8 | 40.6 | 30.1 | 87.0 | 55.9 | 31.1 | 32.3 | 95.7 | 88.4 | 7.3 | 8.5 | 86.5 | 59.1 | 27.5 | 22.3 |
| $\lambda_{\text{fair (mo)}}=100$ | 90.0 | 29.9 | 60.1 | 67.6 | 70.9 | 56.2 | 14.7 | 20.3 | 78.7 | 62.5 | 16.2 | 7.0 | 84.3 | 45.0 | 39.3 | 23.8 | 78.8 | 25.0 | 14.9 | 31.6 | 85.2 | 25.9 | 59.3 | 70.3 | 81.3 | 47.2 | 34.1 | 36.8 |
| $\lambda_{\text{fair (mo)}}=5000$ | 89.6 | 29.0 | 60.6 | 68.5 | 69.4 | 54.6 | 14.7 | 19.5 | 79.9 | 68.1 | 11.8 | 25.0 | 84.3 | 45.0 | 39.3 | 24.2 | 77.4 | 25.0 | 13.8 | 31.6 | 84.7 | 25.9 | 58.8 | 70.6 | 80.9 | 47.7 | 33.2 | 36.4 |
| **FAIRTPT** *special cases without ELRA* | | | | | | | | | | | | | | | | | | | | | | | | | | | | |
| $\lambda_{\text{fair}}=0$ | 93.6 | 85.0 | 8.6 | 12.4 | 92.0 | 42.6 | 49.4 | 37.6 | 59.4 | 20.0 | 39.4 | 25.0 | 84.0 | 40.8 | 43.1 | 37.1 | 87.5 | 57.7 | 29.8 | 26.3 | 94.3 | 86.4 | 7.9 | 8.0 | 85.1 | 55.4 | 29.7 | 21.1 |
| $\lambda_{\text{fair}}=\infty$ | 88.2 | 24.8 | 63.4 | 72.3 | 67.3 | 50.1 | 17.2 | 23.6 | 80.4 | 25.0 | 25.0 | 6.6 | 84.2 | 45.4 | 38.8 | 22.7 | 75.1 | 62.0 | 25.0 | 28.8 | 82.0 | 13.3 | 68.6 | 82.4 | 79.5 | 44.2 | 35.3 | 39.4 |

Table 9: Overall and subgroup-level performance evaluation of FAIRTPT and FAIRTPT (MO) (with and without ELRA). We report mean over 5 random seeds and mean aggregation over all equally sized datasets. Best results are in **bold**, second best underlined. The values that improve upon ZERO-SHOT by more than 2.0 percentage points are highlighted in green, while degradations greater than 2.0 percentage points are shown in red.

| METHOD | FAIRFACE Gender × Race | | | | CELEBA Hair color × Gender | | | | Smiling × Gender | | | | WATERBIRDS Type × Background | | | | UTKFACE Age × Race | | | | Gender × Race | | | | AVERAGE RESULTS | | | |
|---|---|---|---|---|---|---|---|---|---|---|---|---|---|---|---|---|---|---|---|---|---|---|---|---|---|---|---|---|---|
| | A | WGA | B | EOD | A | WGA | B | EOD | A | WGA | B | EOD | A | WGA | B | EOD | A | WGA | B | EOD | A | WGA | B | EOD | A | WGA | B | EOD |
| Zero Shot | 95.7 | 90.0 | 5.7 | 9.4 | 86.4 | 67.8 | 18.6 | 22.3 | 75.8 | 53.7 | 22.1 | 8.0 | 83.8 | 40.2 | 43.7 | 25.0 | 80.3 | 45.7 | 34.5 | 23.2 | 97.1 | 90.1 | 7.0 | 9.1 | 86.5 | 64.6 | 21.9 | 16.2 |
| **FAIRTPT and FAIRTPT (MO)** *with S loss* | | | | | | | | | | | | | | | | | | | | | | | | | | | | |
| $\lambda_{\text{fair}}=1$ | 95.5 | 90.5 | 5.0 | 8.8 | 86.2 | 67.7 | 18.5 | 22.0 | 75.3 | 52.6 | 22.7 | 8.1 | 83.9 | 41.0 | 42.8 | 26.2 | 80.3 | 46.5 | 33.8 | 22.5 | 97.0 | 90.6 | 6.5 | 8.7 | 86.4 | 64.8 | 21.6 | 16.1 |
| $\lambda_{\text{fair}}=100$ | 95.5 | 90.7 | 4.8 | 8.1 | 85.6 | 67.7 | 17.9 | 21.9 | 75.9 | 56.1 | 19.8 | 6.2 | 83.2 | 42.4 | 40.8 | 23.5 | 81.0 | 51.5 | 29.5 | 23.4 | 96.7 | 90.6 | 6.1 | 8.6 | 86.3 | 66.5 | 19.8 | 15.3 |
| $\lambda_{\text{fair}}=5000$ | 95.4 | 90.6 | 4.9 | 8.1 | 85.5 | 67.5 | 18.0 | 21.5 | 75.9 | 55.6 | 20.3 | 5.8 | 83.3 | 43.6 | 39.7 | 23.4 | 80.2 | 49.4 | 30.8 | 26.4 | 96.7 | 90.1 | 6.6 | 9.0 | 86.2 | 66.1 | 20.0 | 15.7 |
| $\lambda_{\text{fair}}=\infty$ | 95.5 | 90.1 | 5.4 | 9.0 | 85.2 | 67.4 | 17.8 | 21.6 | 75.9 | 55.8 | 20.1 | 5.9 | 83.5 | 44.5 | 39.0 | 25.0 | 80.8 | 49.7 | 31.1 | 27.6 | 96.8 | 90.3 | 6.5 | 8.9 | 86.3 | 66.3 | 20.0 | 15.8 |
| $\lambda_{\text{fair (mo)}}=1$ | 95.4 | 90.0 | 5.4 | 9.5 | 86.3 | 67.8 | 18.6 | 22.1 | 75.3 | 52.3 | 23.0 | 8.4 | 83.8 | 40.9 | 42.9 | 26.2 | 80.3 | 46.7 | 33.6 | 22.3 | 97.0 | 90.6 | 6.5 | 8.7 | 86.4 | 64.7 | 21.7 | 16.2 |
| $\lambda_{\text{fair (mo)}}=100$ | 95.3 | 90.4 | 4.9 | 7.9 | 85.3 | 67.5 | 17.8 | 21.3 | 76.1 | 57.8 | 18.3 | 6.3 | 83.2 | 41.4 | 41.8 | 24.6 | 80.8 | 51.6 | 29.3 | 20.9 | 96.6 | 90.9 | 5.8 | 8.1 | 86.2 | 66.6 | 19.6 | 14.8 |
| $\lambda_{\text{fair (mo)}}=5000$ | 95.3 | 90.2 | 5.2 | 8.4 | 85.2 | 67.4 | 17.8 | 22.0 | 76.0 | 56.3 | 19.7 | 6.2 | 83.2 | 41.7 | 41.5 | 24.8 | 80.9 | 48.9 | 32.0 | 26.3 | 96.7 | 90.3 | 6.3 | 8.5 | 86.2 | 65.8 | 20.4 | 16.0 |
| **FAIRTPT and FAIRTPT (MO)** *with TS loss* | | | | | | | | | | | | | | | | | | | | | | | | | | | | |
| $\lambda_{\text{fair}}=1$ | 95.5 | 90.1 | 5.3 | 9.4 | 86.6 | 67.8 | 18.8 | 22.3 | 75.2 | 52.5 | 22.7 | 7.6 | 83.9 | 40.5 | 43.4 | 26.7 | 80.4 | 46.9 | 33.5 | 22.1 | 96.9 | 90.6 | 6.3 | 8.6 | 86.4 | 64.7 | 21.7 | 16.1 |
| $\lambda_{\text{fair}}=100$ | 95.3 | 89.1 | 6.2 | 9.9 | 86.1 | 68.0 | 18.1 | 20.2 | 75.0 | 54.0 | 20.9 | 5.3 | 84.0 | 42.2 | 41.8 | 24.8 | 81.6 | 48.0 | 33.6 | 24.3 | 96.9 | 90.2 | 6.2 | 8.9 | 86.4 | 65.3 | 21.1 | 15.6 |
| $\lambda_{\text{fair}}=5000$ | 95.2 | 89.6 | 5.7 | 9.7 | 86.2 | 67.9 | 18.3 | 20.4 | 75.2 | 55.0 | 20.2 | 6.0 | 84.1 | 42.2 | 41.9 | 24.6 | 81.5 | 47.0 | 34.5 | 24.5 | 96.5 | 90.5 | 6.0 | 8.3 | 86.5 | 65.4 | 21.1 | 15.6 |
| $\lambda_{\text{fair}}=\infty$ | 95.2 | 88.6 | 6.6 | 10.5 | 86.3 | 67.9 | 18.4 | 20.3 | 75.1 | 54.9 | 20.2 | 5.3 | 84.1 | 42.1 | 42.1 | 25.1 | 81.5 | 46.1 | 35.4 | 27.1 | 96.4 | 90.8 | 5.7 | 8.2 | 86.4 | 65.1 | 21.4 | 16.1 |
| $\lambda_{\text{fair (mo)}}=1$ | 95.5 | 90.5 | 5.0 | 9.0 | 86.5 | 67.8 | 18.7 | 22.1 | 75.3 | 52.7 | 22.7 | 7.8 | 84.0 | 40.7 | 43.4 | 26.3 | 80.4 | 46.9 | 33.6 | 22.1 | 96.9 | 90.6 | 6.3 | 8.6 | 86.4 | 64.8 | 21.6 | 16.0 |
| $\lambda_{\text{fair (mo)}}=100$ | 95.0 | 88.0 | 7.1 | 10.9 | 86.1 | 67.9 | 18.1 | 20.2 | 75.0 | 52.6 | 22.4 | 7.1 | 84.1 | 41.4 | 42.7 | 25.6 | 80.9 | 47.0 | 33.9 | 22.8 | 96.3 | 91.0 | 5.2 | 7.3 | 86.2 | 64.7 | 21.6 | 15.6 |
| $\lambda_{\text{fair (mo)}}=5000$ | 94.9 | 88.0 | 6.8 | 10.7 | 86.1 | 67.9 | 18.2 | 20.4 | 75.7 | 55.1 | 20.5 | 5.4 | 84.0 | 41.5 | 42.5 | 25.1 | 80.9 | 45.9 | 34.9 | 24.7 | 96.2 | 90.2 | 5.9 | 8.8 | 86.3 | 64.8 | 21.5 | 15.9 |
| **FAIRTPT and FAIRTPT (MO)** *with Super TS loss* | | | | | | | | | | | | | | | | | | | | | | | | | | | | |
| $\lambda_{\text{fair}}=1$ | 95.7 | 90.5 | 5.2 | 9.0 | 86.5 | 67.7 | 18.8 | 22.4 | 75.3 | 52.4 | 22.9 | 8.2 | 83.9 | 40.0 | 43.9 | 26.1 | 80.3 | 46.3 | 34.0 | 22.7 | 97.1 | 90.6 | 6.5 | 8.6 | 86.5 | 64.6 | 21.9 | 16.2 |
| $\lambda_{\text{fair}}=100$ | 94.8 | 87.7 | 7.1 | 11.7 | 85.7 | 67.9 | 17.7 | 21.1 | 75.2 | 52.8 | 22.4 | 7.5 | 83.5 | 39.3 | 44.2 | 25.5 | 81.1 | 47.8 | 33.3 | 25.0 | 96.5 | 90.6 | 5.9 | 8.3 | 86.1 | 64.4 | 21.8 | 15.7 |
| $\lambda_{\text{fair}}=5000$ | 94.8 | 89.0 | 5.8 | 10.2 | 85.6 | 67.8 | 17.8 | 20.2 | 75.2 | 55.0 | 20.2 | 6.5 | 83.6 | 39.3 | 44.3 | 25.7 | 81.1 | 48.6 | 32.5 | 20.4 | 96.5 | 90.0 | 6.5 | 8.3 | 86.1 | 65.0 | 21.2 | 15.2 |
| $\lambda_{\text{fair}}=\infty$ | 95.0 | 88.3 | 6.7 | 10.2 | 85.7 | 67.9 | 17.8 | 19.8 | 75.5 | 55.9 | 19.6 | 6.5 | 83.5 | 39.5 | 44.0 | 25.5 | 81.1 | 49.1 | 32.0 | 21.3 | 96.4 | 90.3 | 6.1 | 8.4 | 86.2 | 65.2 | 21.0 | 15.3 |
| $\lambda_{\text{fair (mo)}}=1$ | 95.6 | 90.3 | 5.3 | 9.3 | 86.5 | 67.7 | 18.8 | 22.4 | 75.4 | 52.6 | 22.8 | 8.2 | 84.0 | 39.8 | 44.2 | 26.7 | 80.3 | 46.3 | 34.0 | 22.7 | 97.1 | 90.6 | 6.5 | 8.6 | 86.5 | 64.5 | 21.9 | 16.3 |
| $\lambda_{\text{fair (mo)}}=100$ | 95.1 | 88.3 | 6.9 | 9.9 | 85.9 | 67.7 | 18.2 | 20.6 | 75.4 | 52.5 | 22.9 | 8.8 | 83.7 | 39.5 | 44.2 | 25.7 | 80.2 | 46.8 | 33.5 | 22.2 | 96.5 | 91.1 | 5.5 | 7.5 | 86.1 | 64.3 | 21.8 | 15.8 |
| $\lambda_{\text{fair (mo)}}=5000$ | 95.2 | 89.3 | 6.0 | 9.6 | 85.8 | 67.8 | 18.0 | 19.8 | 75.4 | 52.2 | 23.2 | 9.8 | 83.7 | 39.3 | 44.4 | 25.7 | 80.6 | 48.9 | 31.7 | 20.8 | 96.1 | 89.7 | 6.4 | 8.7 | 86.1 | 64.5 | 21.6 | 15.7 |

Table 10: Impact of the loss on the experiment. We report mean over 5 data random seeds and mean aggregation over all equally sized datasets. Best results are in **bold**, second best underlined. The values that improve upon ZERO-SHOT by more than 2.0 percentage points are highlighted in green, while degradations greater than 2.0 percentage points are shown in red.

| METHOD | FAIRFACE | | | | CELEBA | | | | | | | | WATERBIRDS | | | | UTKFACE | | | | | | | | AVERAGE RESULTS | | | |
|---|---|---|---|---|---|---|---|---|---|---|---|---|---|---|---|---|---|---|---|---|---|---|---|---|---|---|---|---|
| | Gender × Race | | | | Hair color × Gender | | | | Smiling × Gender | | | | Type × Background | | | | Age × Race | | | | Gender × Race | | | | | | | |
| | A | WGA | B | EOD | A | WGA | B | EOD | A | WGA | B | EOD | A | WGA | B | EOD | A | WGA | B | EOD | A | WGA | B | EOD | A | WGA | B | EOD |
| ZERO-SHOT | 95.7 | 90.0 | 5.7 | 9.4 | 86.4 | 67.8 | 18.6 | 22.3 | 75.8 | 53.7 | 22.1 | 8.0 | 83.8 | 40.2 | 43.7 | 25.0 | 80.3 | 45.7 | 34.5 | 23.2 | 97.1 | 90.1 | 7.0 | 9.1 | 86.5 | 64.6 | 21.9 | 16.2 |
| *Episodic test-time debiasing methods* | | | | | | | | | | | | | | | | | | | | | | | | | | | | |
| ORTHCALI | 95.9 | 87.8 | 8.1 | 11.3 | 85.1 | 71.6 | 13.5 | 22.0 | 71.4 | 36.8 | 34.6 | 30.6 | 83.3 | 62.0 | 21.4 | 16.2 | 79.8 | 47.0 | 32.8 | 14.4 | 96.8 | 88.9 | 7.9 | 10.1 | 85.4 | 65.7 | 19.7 | 17.4 |
| *Our methods* | | | | | | | | | | | | | | | | | | | | | | | | | | | | |
| FAIRTPT | 95.5 | 90.7 | 4.8 | 8.1 | 85.6 | 67.7 | 17.9 | 21.9 | 75.9 | 56.1 | 19.8 | 6.2 | 83.2 | 42.4 | 40.8 | 23.5 | 81.0 | 51.5 | 29.5 | 23.4 | 96.7 | 90.6 | 6.1 | 8.6 | 86.3 | 66.5 | 19.8 | 15.3 |
| FAIRTPT+ORTHCALI | 95.8 | 90.8 | 5.0 | 8.6 | 85.7 | 67.6 | 18.1 | 23.6 | 76.7 | 54.0 | 22.7 | 17.0 | 83.7 | 42.9 | 40.8 | 17.2 | 81.2 | 50.0 | 31.2 | 23.4 | 96.7 | 90.3 | 6.4 | 8.9 | 86.6 | 65.9 | 20.7 | 16.4 |

Table 11: Overall and subgroup-level performance evaluation of FAIRTPT followed by the orthogonal projection of ORTHCALI (Chuang et al., 2023); we set $\lambda_{\text{fair}} = 100$ and $\lambda_{\text{orth}} = 0$. The spurious embeddings used for projection are those obtained after FAIRTPT. We report mean over 5 random seeds and mean aggregation over all equally sized datasets. Best results are in **bold**, second best underlined. The values that improve upon ZERO-SHOT by more than 2.0 percentage points are highlighted in green, while degradations greater than 2.0 percentage points are shown in red.

| METHOD | FAIRFACE | | | | | | | | | | | | CELEBA | | | | | | | | UTKFACE | | | | | | | | AVERAGE RESULTS | | | |
|---|---|---|---|---|---|---|---|---|---|---|---|---|---|---|---|---|---|---|---|---|---|---|---|---|---|---|---|---|---|---|---|---|
| | Age × Race | | | | Age × Gender | | | | Race × Gender | | | | Makeup × Gender | | | | Glasses × Gender | | | | Age × Gender | | | | Race × Gender | | | | | | | |
| | A | WGA | B | EOD | A | WGA | B | EOD | A | WGA | B | EOD | A | WGA | B | EOD | A | WGA | B | EOD | A | WGA | B | EOD | A | WGA | B | EOD | A | WGA | B | EOD |
| Zero Shot | 83.3 | 40.1 | 43.2 | 34.2 | 83.3 | 52.8 | 30.5 | 8.0 | 57.6 | 2.5 | 55.1 | 7.1 | 66.1 | 26.7 | 39.4 | 38.6 | 94.5 | 78.8 | 15.7 | 7.9 | 80.3 | 45.3 | 35.0 | 12.3 | 62.3 | 14.3 | 47.9 | 14.0 | 75.3 | 37.2 | 38.1 | 17.4 |
| *Episodic test-time adaptation methods* | | | | | | | | | | | | | | | | | | | | | | | | | | | | | | | | |
| TPT | 86.6 | 54.9 | 31.7 | 27.8 | 86.6 | 61.3 | 25.3 | 12.7 | 64.9 | 27.3 | 37.6 | 6.5 | 51.3 | 13.9 | 37.4 | 26.6 | 95.0 | 23.9 | 71.1 | 14.4 | 87.5 | 60.7 | 26.8 | 20.2 | 73.9 | 42.9 | 31.0 | 10.5 | 78.0 | 40.7 | 37.3 | 17.0 |
| TPT *with ELRA* | 83.5 | 41.8 | 41.7 | 33.5 | 83.5 | 53.5 | 30.0 | 8.4 | 57.8 | 2.7 | 55.1 | 7.2 | 65.8 | 26.3 | 39.5 | 38.1 | 95.0 | 75.0 | 20.0 | 6.3 | 80.3 | 45.2 | 35.1 | 12.6 | 62.5 | 14.6 | 47.9 | 14.3 | 75.5 | 37.0 | 38.5 | 17.2 |
| ZERO | | | | | | | | | | | | | | | | | | | | | | | | | | | | | | | | |
| $\rho = 0.1$ | 85.2 | 51.3 | 33.9 | 23.7 | 85.2 | 55.4 | 29.8 | 15.5 | 57.9 | 5.1 | 52.8 | 8.0 | 59.9 | 19.5 | 40.4 | 34.5 | 96.4 | 39.6 | 56.8 | 16.2 | 86.2 | 58.4 | 27.8 | 18.0 | 54.3 | 8.2 | 46.1 | 19.0 | 75.0 | 33.9 | 41.1 | 19.3 |
| $\rho = 0.25$ | 85.5 | 53.1 | 32.4 | 24.9 | 85.5 | 58.2 | 27.4 | 16.2 | 59.0 | 5.3 | 53.6 | 8.4 | 58.1 | 17.6 | 40.5 | 33.6 | 96.8 | 50.8 | 46.0 | 11.7 | 86.6 | 59.1 | 27.5 | 19.9 | 54.3 | 5.3 | 49.0 | 19.4 | 75.1 | 35.6 | 39.5 | 19.2 |
| $\rho = 0.5$ | 85.7 | 53.6 | 32.1 | 27.5 | 85.7 | 60.2 | 25.5 | 15.5 | 59.6 | 4.8 | 54.9 | 8.2 | 57.4 | 16.9 | 40.5 | 33.2 | 96.8 | 48.1 | 48.7 | 18.4 | 87.0 | 60.4 | 26.6 | 20.5 | 54.9 | 4.7 | 50.2 | 19.6 | 75.3 | 35.5 | 39.8 | 20.4 |
| $\rho = 0.75$ | 85.8 | 56.9 | 28.9 | 24.7 | 85.8 | 61.7 | 24.1 | 14.0 | 59.9 | 4.6 | 55.4 | 7.5 | 57.1 | 17.1 | 40.0 | 32.3 | 96.9 | 51.8 | 45.0 | 15.4 | 87.0 | 60.4 | 26.6 | 19.9 | 54.8 | 4.3 | 50.5 | 19.0 | 75.4 | 36.7 | 38.6 | 19.0 |
| $\rho = 1.0$ | 85.9 | 57.0 | 28.9 | 25.9 | 85.9 | 61.8 | 24.1 | 13.8 | 60.0 | 4.4 | 55.6 | 7.2 | 57.5 | 16.9 | 40.6 | 33.3 | 97.0 | 55.5 | 41.4 | 12.4 | 87.1 | 59.8 | 27.3 | 20.8 | 54.9 | 4.2 | 50.7 | 19.1 | 75.5 | 37.1 | 38.4 | 18.9 |
| *Episodic test-time debiasing methods* | | | | | | | | | | | | | | | | | | | | | | | | | | | | | | | | |
| ORTHCALI | | | | | | | | | | | | | | | | | | | | | | | | | | | | | | | | |
| $\lambda_{\text{orth}} = 0.0$ | 81.6 | 43.5 | 38.1 | 43.0 | 83.5 | 53.3 | 30.2 | 8.5 | 57.6 | 2.7 | 54.8 | 7.4 | 75.7 | 43.9 | 31.8 | 37.7 | 93.4 | 79.3 | 14.1 | 9.9 | 81.0 | 45.9 | 35.0 | 13.5 | 63.8 | 15.7 | 48.1 | 14.8 | 76.7 | 40.6 | 36.0 | 19.3 |
| $\lambda_{\text{orth}} = 1.0$ | 81.7 | 44.0 | 37.7 | 42.5 | 83.3 | 55.0 | 28.2 | 7.1 | 57.7 | 2.7 | 54.9 | 7.5 | 75.7 | 43.8 | 31.9 | 37.8 | 93.4 | 79.0 | 14.3 | 9.6 | 80.3 | 46.3 | 34.0 | 12.5 | 63.9 | 16.0 | 47.9 | 14.7 | 76.5 | 41.0 | 35.6 | 18.8 |
| $\lambda_{\text{orth}} = 10.0$ | 81.6 | 42.5 | 39.2 | 44.0 | 82.2 | 52.1 | 30.1 | 14.3 | 57.6 | 2.5 | 55.1 | 7.4 | 75.4 | 43.3 | 32.1 | 38.0 | 93.1 | 79.0 | 14.1 | 10.8 | 76.5 | 37.8 | 38.7 | 12.6 | 64.1 | 18.2 | 45.9 | 13.1 | 75.8 | 39.3 | 36.4 | 20.0 |
| $\lambda_{\text{orth}} = 1000.0$ | 78.2 | 42.6 | 35.6 | 45.1 | 72.2 | 10.5 | 61.7 | 72.0 | 57.7 | 3.3 | 54.3 | 7.7 | 69.5 | 31.8 | 37.8 | 40.4 | 87.2 | 72.8 | 14.5 | 20.0 | 67.8 | 6.4 | 61.4 | 52.2 | 66.8 | 32.8 | 34.0 | 4.3 | 71.3 | 28.6 | 42.7 | 34.5 |
| $\lambda_{\text{orth}} = 100000$ | 76.7 | 44.2 | 32.5 | 45.6 | 71.5 | 9.8 | 61.7 | 73.1 | 57.4 | 2.9 | 54.6 | 7.8 | 68.1 | 29.7 | 38.4 | 40.2 | 86.3 | 72.1 | 14.2 | 21.2 | 67.2 | 6.1 | 61.2 | 53.1 | 67.4 | 34.4 | 33.1 | 3.9 | 70.7 | 28.4 | 42.2 | 35.0 |
| FAIRTPT *and* FAIRTPT (MO) *with S loss and no ELRA* | | | | | | | | | | | | | | | | | | | | | | | | | | | | | | | | |
| $\lambda_{\text{fair}} = 1$ | 85.5 | 57.0 | 28.5 | 26.1 | 85.6 | 61.9 | 23.7 | 14.3 | 63.6 | 20.8 | 42.8 | 6.7 | 57.9 | 17.5 | 40.5 | 33.4 | 97.2 | 56.7 | 40.5 | 12.5 | 86.7 | 58.6 | 28.1 | 21.5 | 73.2 | 39.7 | 33.4 | 12.1 | 78.5 | 44.6 | 33.9 | 18.1 |
| $\lambda_{\text{fair}} = 100$ | 77.1 | 61.7 | 15.4 | 28.3 | 67.3 | 52.9 | 14.4 | 17.2 | 56.4 | 3.3 | 53.1 | 12.1 | 64.4 | 27.7 | 36.7 | 34.5 | 95.3 | 63.9 | 31.4 | 22.1 | 70.3 | 53.0 | 17.3 | 28.3 | 66.3 | 20.4 | 46.0 | 15.7 | 71.0 | 40.4 | 30.6 | 22.6 |
| $\lambda_{\text{fair}} = 5000$ | 75.7 | 60.6 | 15.1 | 26.9 | 65.0 | 49.6 | 15.4 | 17.7 | 56.1 | 3.3 | 52.8 | 12.1 | 64.6 | 29.6 | 35.0 | 32.5 | 94.5 | 60.7 | 33.8 | 25.9 | 68.7 | 51.2 | 17.5 | 28.6 | 66.0 | 20.4 | 45.7 | 15.7 | 70.1 | 39.3 | 30.7 | 22.8 |
| $\lambda_{\text{fair}} = \infty$ | 75.6 | 60.4 | 15.2 | 27.1 | 64.9 | 49.5 | 15.4 | 17.7 | 56.0 | 3.5 | 52.5 | 12.1 | 64.7 | 29.9 | 34.8 | 32.4 | 94.5 | 59.0 | 35.4 | 27.6 | 68.4 | 51.2 | 17.3 | 28.9 | 66.0 | 20.8 | 45.3 | 15.4 | 70.0 | 39.2 | 30.8 | 23.0 |
| $\lambda_{\text{fair (mo)}} = 1$ | 85.6 | 57.0 | 28.6 | 26.1 | 85.6 | 61.3 | 24.3 | 15.0 | 63.6 | 20.5 | 43.1 | 6.9 | 57.9 | 17.2 | 40.6 | 33.5 | 97.1 | 56.1 | 41.0 | 12.4 | 86.7 | 58.5 | 28.2 | 21.7 | 72.8 | 38.7 | 34.1 | 12.4 | 78.5 | 44.2 | 34.3 | 18.3 |
| $\lambda_{\text{fair (mo)}} = 100$ | 78.5 | 62.5 | 16.0 | 27.3 | 73.6 | 63.6 | 10.0 | 14.4 | 58.0 | 5.6 | 52.4 | 10.6 | 64.7 | 25.3 | 39.4 | 37.7 | 96.1 | 63.1 | 32.9 | 21.5 | 74.6 | 52.4 | 22.2 | 33.3 | 68.4 | 22.1 | 46.3 | 16.5 | 73.4 | 42.1 | 31.3 | 23.1 |
| $\lambda_{\text{fair (mo)}} = 5000$ | 77.4 | 61.2 | 16.2 | 28.3 | 71.9 | 60.2 | 11.7 | 15.4 | 57.8 | 5.2 | 52.6 | 10.8 | 64.7 | 26.4 | 38.3 | 36.8 | 95.4 | 63.1 | 32.2 | 22.2 | 72.6 | 51.7 | 21.0 | 34.4 | 68.5 | 22.7 | 45.8 | 16.3 | 72.6 | 41.5 | 31.1 | 23.4 |
| FAIRTPT *and* FAIRTPT (MO) *with TS loss and no ELRA* | | | | | | | | | | | | | | | | | | | | | | | | | | | | | | | | |
| $\lambda_{\text{fair}} = 1$ | 85.2 | 56.4 | 28.8 | 27.5 | 85.6 | 61.3 | 24.3 | 15.3 | 64.0 | 17.8 | 46.3 | 8.4 | 56.6 | 17.4 | 39.2 | 30.9 | 96.9 | 54.6 | 42.3 | 13.0 | 86.9 | 59.0 | 27.9 | 31.7 | 69.9 | 31.7 | 38.3 | 13.4 | 77.9 | 42.6 | 35.3 | 18.4 |
| $\lambda_{\text{fair}} = 100$ | 76.9 | 57.6 | 19.3 | 36.0 | 80.1 | 70.5 | 9.6 | 16.7 | 55.3 | 4.2 | 51.1 | 10.4 | 60.9 | 30.7 | 30.2 | 22.8 | 95.9 | 62.5 | 33.4 | 17.5 | 81.0 | 71.5 | 9.5 | 18.6 | 58.6 | 20.7 | 37.9 | 6.5 | 72.7 | 45.4 | 27.3 | 18.4 |
| $\lambda_{\text{fair}} = 5000$ | 75.8 | 57.6 | 18.2 | 36.2 | 78.8 | 68.3 | 10.5 | 17.2 | 54.2 | 3.5 | 50.7 | 11.8 | 60.8 | 32.1 | 28.7 | 21.9 | 95.0 | 64.4 | 30.6 | 17.1 | 79.5 | 70.8 | 8.7 | 18.6 | 58.1 | 21.3 | 36.8 | 6.9 | 71.7 | 45.4 | 26.3 | 18.5 |
| $\lambda_{\text{fair}} = \infty$ | 75.8 | 57.6 | 18.1 | 36.2 | 78.7 | 68.1 | 10.6 | 17.4 | 54.2 | 3.5 | 50.7 | 11.7 | 60.8 | 32.3 | 28.5 | 21.7 | 94.9 | 64.0 | 30.9 | 17.4 | 79.7 | 71.0 | 8.7 | 18.5 | 58.2 | 21.2 | 37.0 | 7.0 | 71.8 | 45.4 | 26.4 | 18.6 |
| $\lambda_{\text{fair (mo)}} = 1$ | 85.2 | 56.4 | 28.8 | 28.3 | 85.7 | 61.5 | 24.2 | 14.8 | 63.9 | 17.1 | 46.8 | 8.6 | 56.2 | 16.9 | 39.3 | 30.9 | 96.9 | 54.6 | 42.3 | 13.0 | 87.0 | 59.0 | 27.9 | 20.4 | 69.9 | 31.4 | 38.6 | 13.6 | 77.8 | 42.4 | 35.4 | 18.5 |
| $\lambda_{\text{fair (mo)}} = 100$ | 78.9 | 55.0 | 23.8 | 38.4 | 81.9 | 75.3 | 6.5 | 12.7 | 58.5 | 7.5 | 51.0 | 8.9 | 58.9 | 26.6 | 32.3 | 23.7 | 96.6 | 62.3 | 34.2 | 15.9 | 81.8 | 71.3 | 10.5 | 16.4 | 62.0 | 21.0 | 41.6 | 9.2 | 74.1 | 45.3 | 28.4 | 18.2 |
| $\lambda_{\text{fair (mo)}} = 5000$ | 78.1 | 57.3 | 20.7 | 35.9 | 81.2 | 73.8 | 7.3 | 13.4 | 58.1 | 7.5 | 50.6 | 8.7 | 59.1 | 27.6 | 31.5 | 23.5 | 95.6 | 61.9 | 33.7 | 19.6 | 80.7 | 71.7 | 9.0 | 16.2 | 62.5 | 21.4 | 41.2 | 9.0 | 73.6 | 45.9 | 27.7 | 18.0 |
| FAIRTPT *and* FAIRTPT (MO) *with Super TS loss and no ELRA* | | | | | | | | | | | | | | | | | | | | | | | | | | | | | | | | |
| $\lambda_{\text{fair}} = 1$ | 86.0 | 56.8 | 29.2 | 23.4 | 85.9 | 59.7 | 26.2 | 16.4 | 61.4 | 10.0 | 51.4 | 7.0 | 55.7 | 16.3 | 39.4 | 30.8 | 96.8 | 51.2 | 45.6 | 15.6 | 87.0 | 59.0 | 28.0 | 21.0 | 59.8 | 12.4 | 47.3 | 17.2 | 76.1 | 37.9 | 38.2 | 18.8 |
| $\lambda_{\text{fair}} = 100$ | 74.6 | 45.6 | 29.0 | 41.5 | 80.6 | 71.5 | 9.2 | 17.4 | 56.7 | 6.5 | 50.3 | 20.4 | 56.7 | 25.9 | 30.7 | 21.4 | 81.5 | 22.1 | 59.4 | 48.7 | 82.9 | 66.7 | 16.2 | 11.5 | 68.3 | 11.6 | 56.6 | 27.0 | 71.6 | 35.7 | 35.9 | 26.8 |
| $\lambda_{\text{fair}} = 5000$ | 72.0 | 43.6 | 28.4 | 47.1 | 77.2 | 65.0 | 12.2 | 21.2 | 50.2 | 4.5 | 45.7 | 25.1 | 56.5 | 27.4 | 29.1 | 28.0 | 68.4 | 15.8 | 52.7 | 62.6 | 79.8 | 65.9 | 13.9 | 12.0 | 64.6 | 9.0 | 55.6 | 32.7 | 67.0 | 33.0 | 33.9 | 32.7 |
| $\lambda_{\text{fair}} = \infty$ | 71.9 | 43.5 | 28.3 | 46.5 | 77.0 | 64.7 | 12.4 | 21.4 | 49.7 | 4.0 | 45.7 | 25.8 | 56.3 | 27.3 | 29.1 | 27.9 | 68.0 | 15.4 | 52.5 | 62.9 | 79.8 | 66.3 | 13.5 | 12.5 | 64.4 | 9.1 | 55.3 | 33.1 | 66.7 | 32.9 | 33.8 | 32.9 |
| $\lambda_{\text{fair (mo)}} = 1$ | 86.0 | 56.2 | 29.7 | 24.0 | 85.9 | 59.8 | 26.1 | 16.3 | 61.4 | 9.8 | 51.5 | 7.1 | 55.6 | 15.9 | 39.7 | 31.2 | 96.9 | 51.2 | 45.6 | 15.6 | 87.0 | 58.8 | 28.2 | 21.2 | 59.8 | 12.3 | 47.5 | 17.2 | 76.1 | 37.7 | 38.3 | 18.9 |
| $\lambda_{\text{fair (mo)}} = 100$ | 81.8 | 47.3 | 34.5 | 29.8 | 84.1 | 74.4 | 9.7 | 12.2 | 59.2 | 17.2 | 42.0 | 13.4 | 51.8 | 16.8 | 35.0 | 22.6 | 88.7 | 24.1 | 64.6 | 46.6 | 84.0 | 64.0 | 20.1 | 9.2 | 63.8 | 29.6 | 34.1 | 10.8 | 73.4 | 39.1 | 34.3 | 20.7 |
| $\lambda_{\text{fair (mo)}} = 5000$ | 79.9 | 44.8 | 35.1 | 31.5 | 81.7 | 74.9 | 6.8 | 11.4 | 56.8 | 18.0 | 38.8 | 17.2 | 51.2 | 17.2 | 34.0 | 21.3 | 77.3 | 16.9 | 60.4 | 63.1 | 82.3 | 65.8 | 16.5 | 7.5 | 63.6 | 26.4 | 37.2 | 12.4 | 70.4 | 37.7 | 32.7 | 23.5 |
| FAIRTPT *and* FAIRTPT (MO) *with S loss* | | | | | | | | | | | | | | | | | | | | | | | | | | | | | | | | |
| $\lambda_{\text{fair}} = 1$ | 83.5 | 44.2 | 39.3 | 32.0 | 83.4 | 53.9 | 29.5 | 8.2 | 57.7 | 2.7 | 55.0 | 7.3 | 66.0 | 26.3 | 39.6 | 38.6 | 94.9 | 78.2 | 16.0 | 7.8 | 80.4 | 45.7 | 34.7 | 12.3 | 62.4 | 14.3 | 48.1 | 14.5 | 75.5 | 38.0 | 37.5 | 17.2 |
| $\lambda_{\text{fair}} = 100$ | 83.0 | 50.0 | 33.0 | 31.0 | 83.1 | 60.0 | 23.1 | 7.7 | 57.2 | 4.0 | 53.2 | 6.8 | 67.1 | 27.6 | 39.3 | 39.1 | 94.6 | 78.2 | 16.4 | 7.6 | 80.0 | 49.5 | 30.5 | 12.6 | 62.2 | 15.8 | 46.4 | 13.6 | 75.3 | 40.8 | 34.6 | 16.9 |
| $\lambda_{\text{fair}} = 5000$ | 82.7 | 50.4 | 32.3 | 29.1 | 83.3 | 62.7 | 20.6 | 6.6 | 57.2 | 3.8 | 53.4 | 7.2 | 67.1 | 27.6 | 39.5 | 39.4 | 94.6 | 75.4 | 19.2 | 11.5 | 80.2 | 49.2 | 31.0 | 12.6 | 62.0 | 15.4 | 46.5 | 13.6 | 75.3 | 40.7 | 34.6 | 17.1 |
| $\lambda_{\text{fair}} = \infty$ | 82.7 | 48.9 | 33.8 | 31.0 | 83.2 | 63.6 | 19.6 | 5.4 | 57.4 | 3.8 | 53.6 | 7.4 | 67.3 | 29.8 | 39.9 | 40.4 | 94.3 | 74.3 | 20.4 | 11.9 | 80.1 | 49.2 | 30.9 | 12.4 | 62.0 | 15.4 | 46.7 | 13.5 | 75.3 | 40.4 | 35.0 | 17.4 |
| $\lambda_{\text{fair (mo)}} = 1$ | 83.5 | 43.6 | 39.9 | 32.2 | 83.4 | 53.7 | 29.7 | 8.2 | 57.7 | 2.7 | 55.0 | 7.3 | 65.9 | 26.1 | 39.8 | 38.8 | 94.9 | 78.6 | 16.3 | 8.1 | 80.3 | 45.5 | 34.8 | 12.3 | 62.5 | 14.3 | 48.1 | 14.5 | 75.5 | 37.8 | 37.7 | 17.3 |
| $\lambda_{\text{fair (mo)}} = 100$ | 83.4 | 49.0 | 34.4 | 30.5 | 83.0 | 60.0 | 23.0 | 7.7 | 57.2 | 3.9 | 53.3 | 7.1 | 67.1 | 28.0 | 39.1 | 38.4 | 94.8 | 77.5 | 17.4 | 7.8 | 80.2 | 51.8 | 28.5 | 13.3 | 62.3 | 18.3 | 44.9 | 12.9 | 75.6 | 41.2 | 34.4 | 16.6 |
| $\lambda_{\text{fair (mo)}} = 5000$ | 83.0 | 48.8 | 34.2 | 28.9 | 82.6 | 59.3 | 23.3 | 7.6 | 57.3 | 4.1 | 53.2 | 6.7 | 67.1 | 27.8 | 39.3 | 39.7 | 94.6 | 78.5 | 16.1 | 7.9 | 80.0 | 50.1 | 29.9 | 12.6 | 63.0 | 18.4 | 44.6 | 12.5 | 75.4 | 41.0 | 34.4 | 16.6 |
| FAIRTPT *and* FAIRTPT (MO) *with TS loss* | | | | | | | | | | | | | | | | | | | | | | | | | | | | | | | | |
| $\lambda_{\text{fair}} = 1$ | 83.6 | 43.3 | 40.3 | 32.5 | 83.6 | 53.7 | 29.9 | 8.8 | 57.7 | 2.7 | 55.0 | 7.3 | 65.7 | 26.1 | 39.6 | 38.4 | 94.8 | 78.2 | 16.6 | 8.2 | 80.4 | 45.8 | 34.6 | 11.7 | 62.5 | 14.3 | 48.2 | 14.5 | 75.5 | 37.7 | 37.7 | 17.3 |
| $\lambda_{\text{fair}} = 100$ | 83.9 | 44.3 | 39.5 | 33.6 | 83.6 | 60.5 | 23.0 | 6.0 | 57.4 | 2.7 | 54.7 | 6.8 | 65.5 | 26.7 | 38.8 | 37.4 | 94.7 | 77.9 | 16.3 | 8.7 | 80.5 | 50.6 | 29.9 | 8.4 | 61.7 | 14.7 | 47.3 | 13.8 | 75.2 | 39.6 | 35.7 | 16.4 |
| $\lambda_{\text{fair}} = 5000$ | 83.6 | 45.9 | 37.7 | 29.4 | 83.6 | 60.5 | 22.9 | 6.4 | 57.4 | 2.9 | 54.6 | 6.8 | 65.6 | 26.5 | 39.1 | 38.1 | 94.5 | 79.2 | 15.4 | 8.0 | 80.5 | 51.1 | 29.4 | 8.8 | 61.8 | 14.9 | 46.9 | 13.4 | 75.3 | 40.1 | 35.1 | 15.8 |
| $\lambda_{\text{fair}} = \infty$ | 83.2 | 43.8 | 39.4 | 31.2 | 83.6 | 61.4 | 22.1 | 5.8 | 57.4 | 2.9 | 54.4 | 6.9 | 65.6 | 26.9 | 38.9 | 37.5 | 94.2 | 77.4 | 16.9 | 8.9 | 80.6 | 51.5 | 29.1 | 8.0 | 61.9 | 15.0 | 46.2 | 14.1 | 75.3 | 39.8 | 35.5 | 16.0 |
| $\lambda_{\text{fair (mo)}} = 1$ | 83.6 | 43.8 | 39.8 | 32.0 | 83.5 | 53.7 | 29.8 | 8.5 | 57.7 | 2.7 | 55.0 | 7.3 | 65.5 | 25.6 | 39.9 | 38.7 | 94.8 | 78.2 | 16.6 | 8.2 | 80.5 | 45.6 | 34.9 | 11.9 | 62.4 | 14.1 | 48.3 | 14.5 | 75.4 | 37.7 | 37.8 | 17.3 |
| $\lambda_{\text{fair (mo)}} = 100$ | 83.2 | 43.3 | 39.9 | 33.7 | 83.4 | 61.7 | 21.6 | 5.8 | 57.6 | 4.2 | 53.3 | 7.0 | 64.7 | 25.8 | 38.9 | 37.3 | 94.7 | 75.8 | 18.9 | 9.7 | 80.8 | 51.9 | 28.9 | 7.7 | 62.9 | 16.2 | 46.2 | 13.7 | 75.3 | 39.9 | 35.4 | 16.3 |
| $\lambda_{\text{fair (mo)}} = 5000$ | 83.3 | 44.0 | 39.3 | 32.2 | 83.6 | 61.3 | 22.2 | 5.4 | 57.7 | 4.8 | 52.9 | 7.1 | 64.6 | 26.0 | 38.6 | 36.7 | 94.4 | 76.7 | 17.7 | 10.1 | 80.7 | 51.4 | 29.3 | 7.8 | 62.8 | 16.4 | 46.4 | 14.2 | 75.3 | 40.1 | 35.2 | 16.2 |
| FAIRTPT *and* FAIRTPT (MO) *with Super TS loss* | | | | | | | | | | | | | | | | | | | | | | | | | | | | | | | | |
| $\lambda_{\text{fair}} = 1$ | 83.5 | 43.2 | 40.3 | 32.3 | 83.5 | 53.3 | 30.2 | 8.8 | 57.9 | 2.5 | 55.3 | 7.2 | 65.7 | 26.1 | 39.6 | 38.3 | 94.9 | 78.5 | 16.4 | 8.1 | 80.4 | 45.5 | 34.9 | 12.2 | 62.2 | 13.7 | 48.5 | 14.4 | 75.4 | 37.5 | 37.9 | 17.3 |
| $\lambda_{\text{fair}} = 100$ | 83.6 | 44.9 | 38.6 | 30.6 | 83.5 | 58.6 | 24.9 | 8.0 | 57.6 | 6.6 | 51.0 | 6.9 | 65.0 | 26.2 | 38.8 | 36.7 | 93.9 | 76.3 | 17.6 | 9.7 | 80.8 | 50.0 | 30.8 | 9.5 | 62.5 | 20.9 | 41.6 | 9.4 | 75.3 | 40.5 | 34.8 | 15.8 |
| $\lambda_{\text{fair}} = 5000$ | 83.3 | 43.2 | 40.1 | 32.5 | 83.5 | 59.6 | 23.9 | 7.8 | 57.5 | 7.0 | 50.4 | 6.8 | 65.4 | 27.2 | 38.2 | 36.5 | 93.9 | 76.9 | 17.0 | 8.6 | 80.7 | 49.7 | 31.1 | 9.1 | 62.3 | 21.4 | 41.0 | 8.4 | 75.2 | 40.7 | 34.5 | 15.6 |
| $\lambda_{\text{fair}} = \infty$ | 83.3 | 45.0 | 38.4 | 28.6 | 83.4 | 58.8 | 24.6 | 7.9 | 57.6 | 6.9 | 50.7 | 6.7 | 65.2 | 27.3 | 37.9 | 35.7 | 93.9 | 76.4 | 17.5 | 9.2 | 80.9 | 50.4 | 30.5 | 8.9 | 62.5 | 21.1 | 41.4 | 8.8 | 75.3 | 40.8 | 34.4 | 15.1 |
| $\lambda_{\text{fair (mo)}} = 1$ | 83.4 | 43.2 | 40.3 | 31.1 | 83.6 | 53.5 | 30.1 | 8.7 | 57.9 | 2.5 | 55.3 | 7.2 | 65.8 | 26.0 | 39.7 | 38.3 | 94.9 | 78.5 | 16.4 | 8.1 | 80.4 | 45.2 | 35.2 | 12.7 | 62.1 | 13.5 | 48.6 | 14.5 | 75.4 | 37.5 | 38.0 | 17.3 |
| $\lambda_{\text{fair (mo)}} = 100$ | 83.5 | 43.9 | 39.7 | 30.7 | 83.8 | 59.5 | 24.4 | 7.3 | 57.7 | 2.9 | 54.8 | 7.2 | 64.6 | 25.0 | 39.6 | 37.8 | 94.0 | 76.4 | 17.6 | 9.6 | 80.8 | 48.8 | 31.9 | 9.6 | 62.1 | 16.4 | 45.7 | 12.6 | 75.2 | 39.0 | 36.2 | 16.4 |
| $\lambda_{\text{fair (mo)}} = 5000$ | 83.5 | 43.7 | 39.8 | 31.3 | 83.8 | 60.0 | 23.8 | 7.1 | 57.7 | 3.6 | 54.1 | 7.4 | 64.3 | 24.3 | 39.9 | 37.8 | 93.9 | 78.9 | 15.0 | 7.7 | 80.6 | 49.1 | 31.5 | 9.4 | 62.0 | 17.0 | 45.0 | 11.7 | 75.1 | 39.5 | 35.6 | 16.0 |

Table 12: Overall and subgroup-level performance evaluation of all considered methods on additional dataset-attribute configurations. We report mean over 5 random seeds and mean aggregation over all equally sized datasets. Best results are in **bold**, second best underlined. The values that improve upon ZERO-SHOT by more than 2.0 percentage points are highlighted in green, while degradations greater than 2.0 percentage points are shown in red.

## A.4 Additional Discussion on FairTPT

**Effectiveness Validation via ASI and ATC Metrics.** To evaluate the fairness and accuracy trade-off introduced by FairTPT, we report two metrics:

- Average Sensitive Indifference (ASI): The average normalized entropy of sensitive attribute predictions over the dataset. A higher ASI indicates reduced reliance on sensitive attributes.

- Average Target Confidence (ATC): Defined as $1-$ average normalized entropy of target attribute predictions over the dataset. A higher ATC reflects greater confidence in target predictions.

Table 13 and Table 14 present ASI and ATC values for all considered methods, averaged over five random seeds and aggregated across equally sized datasets. For FairTPT and FairTPT (MO), we set $\lambda_{\text{fair}} = 100$ and $\lambda_{\text{fair (mo)}} = 100$, respectively. Hyperparameter details for all methods are provided in Table 3. We observe that FairTPT and FairTPT (MO) significantly increase ASI with minimal impact on ATC.

| Dataset | Before Update | FairTPT | FairTPT (MO) | TPT |
|---|---|---|---|---|
| FairFace (Gender × Race) | 42.1 | 53.9 | 57.4 | 39.5 |
| CelebA (Hair color × Gender) | 15.3 | 41.4 | 46.4 | 14.3 |
| CelebA (Smiling × Gender) | 15.3 | 40.1 | 50.9 | 11.9 |
| WaterBirds (Type × Background) | 56.0 | 70.9 | 71.1 | 51.2 |
| UTKFace (Age × Race) | 45.4 | 53.3 | 56.1 | 35.3 |
| UTKFace (Gender × Race) | 45.4 | 55.6 | 57.2 | 29.9 |
| **Average** | 36.6 | 52.5 | 56.5 | 30.3 |

Table 13: ASI evaluation across datasets. Mean over 5 seeds and aggregated results.

| Dataset | Before Update | FairTPT | FairTPT (MO) | TPT |
|---|---|---|---|---|
| FairFace (Gender × Race) | 66.4 | 65.0 | 64.4 | 88.4 |
| CelebA (Hair color × Gender) | 57.5 | 53.9 | 55.1 | 96.0 |
| CelebA (Smiling × Gender) | 45.2 | 44.2 | 43.4 | 80.2 |
| WaterBirds (Type × Background) | 62.1 | 59.7 | 60.7 | 80.0 |
| UTKFace (Age × Race) | 46.1 | 45.2 | 44.8 | 85.0 |
| UTKFace (Gender × Race) | 71.6 | 70.6 | 69.3 | 91.4 |
| **Average** | 58.1 | 56.4 | 56.3 | 86.8 |

Table 14: ATC evaluation across datasets. Mean over 5 seeds and aggregated results.

**Hyperparameter Tuning and Runtime Analysis.** FairTPT exhibits inference-time cost comparable to TPT (prompt-tuning strategies). Table 15 reports runtime per image (in seconds) under identical hardware for FairFace (Gender × Race): OrthCali requires solving an optimization

| Method | TPT | Zero | FairTPT | FairTPT (MO) |
|---|---|---|---|---|
| Runtime (sec) | 0.43 | 0.04 | 1.03 | 1.71 |

Table 15: Inference-time runtime per image (sec) on FairFace (Gender × Race).

problem upfront, making its inference-time cost closer to zero-shot. The reported FairTPT runtime includes automatic learning-rate adaptation (ELRA). Unlike OrthCali, which requires tuning $\lambda_{\text{orth}}$ using a *labeled* validation set (contrary to the unsupervised test-time setting), FairTPT is robust to $\lambda_{\text{fair}}$ and does not require additional tuning.

**Support for Multiple Sensitive Attributes.** FAIRTPT is not restricted to a single sensitive attribute and supports multiple attributes via:

- Independent Treatment: Extend the optimization objective in Equation 3 by adding spurious entropy terms for each sensitive attribute (e.g., age, gender, race).
- Joint Treatment: Define a joint sensitive attribute over the product space (e.g., age $\times$ gender $\times$ race) and apply Equation 3 directly.

As a test-time debiasing approach, FAIRTPT allows the fairness auditor to specify sensitive attributes at inference. For an unlabeled test image, the user can select attributes and their possible values for debiasing. If gender is specified, debiasing applies only to gender. The method does not automatically detect spurious attributes but can incorporate user-provided or externally inferred factors (e.g., via GPT-based tools).

## B   SCALING STEPS IN THE PROBABILITY SIMPLEX

Given our rescaling of the learning rate, this section investigates the relationship between changes in a softmax probability vector and changes in its leading ($\arg\max$) component. We establish the following result:

**Proposition 1.** *Let $\boldsymbol{p}$ be a probability vector in the set $\mathcal{D}_i = \{\boldsymbol{x} \in S_n \mid x_i \geq x_j, \forall j \neq i\}$, where $S_n$ denotes the $n$-dimensional probability simplex. We define the distance function $d : S_n^2 \to \mathbb{R}_+$ by*

$$d(\boldsymbol{p}, \boldsymbol{q}) := \sum_i \max(0, p_i - q_i) = \frac{1}{2} \sum_i |p_i - q_i|,$$

*and the index mapping $I : S_n \to [n]$ (where $[n] = \{1, \ldots, n\}$) by $I(\boldsymbol{p}) = \min(\arg\max_i p_i)$. Then, for all $(\boldsymbol{p}, \boldsymbol{q}) \in S_n^2$, the following hold:*

i) $I(\boldsymbol{p}) \neq I(\boldsymbol{q}) \implies d(\boldsymbol{p}, \boldsymbol{q}) > \frac{1}{2}(\max_i p_i - \max_{j \neq I(\boldsymbol{p})} p_j)$

ii) $2d(\boldsymbol{p}, \boldsymbol{q}) > \displaystyle\max_{\Omega \subset [n] \setminus \{I(\boldsymbol{p})\}} \left| p_1 - \frac{1}{1+|\Omega|} \right| + \sum_{i=2}^n \left| p_i - \frac{1}{1+|\Omega|} \sum_{j \in \Omega} \delta_{ij} \right| \implies I(\boldsymbol{p}) \neq I(\boldsymbol{q})$

   *(where $\delta_{ij} = 1$ if $i = j$ and $0$ otherwise)*

iii) $d(\boldsymbol{p}, \boldsymbol{q}) > \left(1 - \frac{1}{n}\right) \implies I(\boldsymbol{p}) \neq I(\boldsymbol{q})$

*Proof.* We first note that $d$ is a metric on $S_n$ since it satisfies non-negativity, symmetry, and the triangle inequality. In fact, $d$ is convex in both arguments. Without loss of generality, assume $\boldsymbol{p} \in \mathcal{D}_1$.

i) We seek the minimal change in $\boldsymbol{p}$ required to leave $\mathcal{D}_1$:

$$\inf_{\boldsymbol{q} \in S_n \setminus \mathcal{D}_1} d(\boldsymbol{p}, \boldsymbol{q}).$$

Since $d$ is half the $\ell_1$ distance, the smallest perturbation that changes the leading index reduces $p_1$ and increases the second-largest component equally. This yields

$$\frac{1}{2}(p_1 - \max_{i>1} p_i).$$

In the case $n = 2$, this reduces to $p_1 - \frac{1}{2}$.

ii) We next determine the largest distance within $\mathcal{D}_1$ from $\mathbf{p}$:

$$\sup_{\boldsymbol{w} \in \mathcal{D}_1} d(\boldsymbol{p}, \boldsymbol{w}).$$

Since $\mathcal{D}_1$ is convex and $d$ is convex in each argument, the supremum is attained at an extreme point of $\mathcal{D}_1$:

$$\sup_{\boldsymbol{w} \in \mathcal{D}_1} d(\boldsymbol{p}, \boldsymbol{w}) = \sup_{\boldsymbol{w} \in \text{ext}(\mathcal{D}_1)} d(\boldsymbol{p}, \boldsymbol{w}).$$

The extreme points have the form

$$\left( \frac{1}{1+|\Omega|}, a_2, \ldots, a_n \right),$$

where $\Omega \subset \{2, \ldots, n\}$ and $a_i = \frac{1}{1+|\Omega|}$ if $i \in \Omega$, and $a_i = 0$ otherwise. This yields

$$\max_{\boldsymbol{w} \in \text{ext}(\mathcal{D}_1)} 2d(\boldsymbol{p}, \boldsymbol{w}) = \max_{\Omega \subset [n] \setminus [1]} \left| p_1 - \frac{1}{1+|\Omega|} \right| + \sum_{i=2}^n \left| p_i - \frac{1}{1+|\Omega|} \sum_{j \in \Omega} \delta_{ij} \right|.$$

For $n = 2$, this simplifies to

$$\max_{\boldsymbol{w} \in \text{ext}(\mathcal{D}_1)} d(\boldsymbol{p}, \boldsymbol{w}) = \frac{1}{2} \max \left\{ |p_1 - 1| + |1 - p_1|, \left| p_1 - \frac{1}{2} \right| + \left| 1 - p_1 - \frac{1}{2} \right| \right\}$$

$$= (1 - p_1) \mathbb{1}\left[ p_1 \leq \frac{3}{4} \right] + \left( p_1 - \frac{1}{2} \right) \mathbb{1}\left[ p_1 > \frac{3}{4} \right].$$

iii) Finally, to obtain a bound independent of $p$, we compute

$$\sup_{(\boldsymbol{p},\boldsymbol{w})\in\mathcal{D}_1^2} d(\boldsymbol{p},\boldsymbol{w}).$$

By convexity of $\mathcal{D}_1$ and $d$, this supremum is attained at extreme points:

$$\sup_{(\boldsymbol{p},\boldsymbol{w})\in\mathcal{D}_1^2} d(\boldsymbol{p},\boldsymbol{w}) = \max_{(\boldsymbol{p},\boldsymbol{w})\in\text{ext}(\mathcal{D}_1)^2} d(\boldsymbol{p},\boldsymbol{w}).$$

Given the definition of $d$, the maximum distance is attained for a pair of vectors of the form $(\frac{1}{m_1},\ldots,\frac{1}{m_1},0,\ldots,0)$ and $(\frac{1}{m_2},0,\ldots,0,\frac{1}{m_2},\ldots,\frac{1}{m_2})$, where $1 \leq m_1, m_2 \leq n$ and $m_1 \leq m_2$ without loss of generality. The associated distance is $1 - \frac{1}{m_2}$, which is maximized when $m_2 = n$. This forces $m_1 = 1$, recovering the known result that the maximum is attained between the vertices $(1,0,\ldots,0)$ and $(\frac{1}{n},\ldots,\frac{1}{n})$ vertices. Hence,

$$\sup_{(\boldsymbol{p},\boldsymbol{w})\in\mathcal{D}_1^2} d(\boldsymbol{p},\boldsymbol{w}) = \left(1 - \frac{1}{n}\right).$$

For $n = 2$, this equals $\frac{1}{2}$, and for large $n$, it approaches 1.

$\square$

These results suggest the following:

1. Statement (i) indicates that a measure of the initial confidence provides an upper bound on indifference, while (ii) and (iii) yield lower bounds for collapse. The learning rate must therefore be chosen carefully, taking into account both the input and the step size, to avoid either collapse or indifference.

2. A lower learning rate increases the likelihood of accuracy changes for the least confident samples, i.e., those lying close to vertices with at least two similar probabilities.

We now compute explicitly the typical distance covered by one optimizer step in the linear regime. The parameter update is $\Delta\boldsymbol{t}_{\text{ctx}} = -\eta\nabla_{\boldsymbol{t}_{\text{ctx}}}\ell$, where $\nabla_{\boldsymbol{t}_{\text{ctx}}}\ell = w_1\nabla_{\boldsymbol{t}_{\text{ctx}}}\ell_{\text{ent}}(x,\{\boldsymbol{t}_{\text{ctx}};\mathcal{Y}\}) + w_2 g^{\mathcal{S}}_{\boldsymbol{t}_{\text{ctx}}}$ with $w_1, w_2$ normalized weights given by our method or UPGrad aggregation, and $g^{\mathcal{S}}_{\boldsymbol{t}_{\text{ctx}}}$ the spurious entropy contribution. We recall that

$$\nabla_{\boldsymbol{t}_{\text{ctx}}}\ell_{\text{ent}}(x,\{\boldsymbol{t}_{\text{ctx}};\mathcal{Y}\}) = -\sum_i (1 + \log\bar{p}_i)\nabla_{\boldsymbol{t}_{\text{ctx}}}\bar{p}_i,$$

which, in the linear regime and in terms of $\boldsymbol{p}$, leads to

$$d\left(\boldsymbol{p}(\boldsymbol{t}_{\text{ctx}}),\boldsymbol{p}(\boldsymbol{t}_{\text{ctx}} + \Delta\boldsymbol{t}_{\text{ctx}})\right) \simeq \frac{1}{2}\eta\sum_i |\nabla_{\boldsymbol{t}_{\text{ctx}}}l \cdot \nabla_{\boldsymbol{t}_{\text{ctx}}}p_i|$$

$$= \frac{1}{2}\eta\sum_i \left| -w_1\sum_j (1 + \log\bar{p}_j)\nabla_{\boldsymbol{t}_{\text{ctx}}}\bar{p}_j \cdot \nabla_{\boldsymbol{t}_{\text{ctx}}}p_i + w_2 g^{\mathcal{S}}_{\boldsymbol{t}_{\text{ctx}}} \cdot \nabla_{\boldsymbol{t}_{\text{ctx}}}p_i \right|.$$

This expression highlights the high dependency of the distance, and so of an appropriate choice of $\eta$, on the initial softmax output. For a given $\boldsymbol{p}$, there is no straightforward way to choose $\eta$ to yield the same distance across all steps and inputs. Setting $\eta$ via ELRA to ensure a consistent change in the target marginal entropy is thus a heuristic approach. In the linear approximation,

$$\beta := 0.01 \simeq \eta \left| \nabla_{\boldsymbol{t}_{\text{ctx}}}\ell \cdot \sum_i (1 + \log\bar{p}_i)\nabla_{\boldsymbol{t}_{\text{ctx}}}\bar{p}_i \right|,$$

motivated by the direct link between confidence (measured by entropy) and accuracy in a calibrated model.

