# OpenReview forum: "Fairness-Aware Test-Time Prompt Tuning"
_ICLR.cc/2026/Conference — Submitted to ICLR 2026_

### Official Review · Reviewer_YdNt · 2025-10-25

**Soundness:** 3
**Presentation:** 2
**Contribution:** 2
**Rating:** 6
**Confidence:** 3

**Summary:**

This paper propose a fairness-aware test-time prompt tuning method by minimizing the entropy of target predictions while reducing bias by maximizing the entropy of sensitive attribute predictions. It achieves balanced adaptation through a lightweight learning-rate heuristic that prevents over-debiasing, all in a fully unsupervised, episodic setting without needing sensitive attribute labels.

**Strengths:**

- The motivation of this paper is clear, and the objective function (min target entropy + max spurious entropy) is very intuitive and easy to understand.

- The method does not require access to sensitive attributes or retraining the model and it's easy and practical for deployed models since it only injects in the test stage through soft prompting.

- The empirical evaluation covered most of the benchmark datasets in fairness for image classification tasks, which makes the results rigorous.

**Weaknesses:**

- The method is limited on one attribute spurious correlation, not extend to complex or multi-sensitive attributes, for example, if the spurious correlation is on a combination of age, gender and race. Also, the method depends on pre-define the sensitive attribute and in practice, sometimes it may not reflect the true spurious correlation.

- Lack of analysis of explainability, for example, using attention map to illustrate the change of sensitive attribute after applying the method (is it really reduce the dependence of sensitive attribute).

- No ablation study about if remove ELRA or view filtering or simply maximising spurious entropy would work, lack of demonstrate of each component of the objective function.

**Questions:**

- If predefined sensitive attribute is gender -> y, the actual spurious is gender+race -> y or race -> y, under such situation, is the method still effective? For example, would the soft prompting still successfully reduce the model's reliance on the true spurious factors, or might it fail to capture the intertwined biases and even introduce unintended distortions in the predictions?

- Does there exist a generalizable way to tune $\lambda_{fair}$ across datasets or tasks, and how sensitive the model’s performance is to $\lambda_{fair}$

---

> ### Author Response · Authors · 2025-11-19
>
> We thank the reviewer for the thoughtful comments. Below, we address the concerns raised.
>
> ---
>
> **Q1: The method is limited to a single sensitive attribute ... reliance on predefined attributes ... impact when the true spurious attribute is different ...**
>
> FairTPT is not limited to a single sensitive attribute. It supports multiple sensitive attributes in two ways:
> - Independent treatment: In the optimization objective (Eq. 3), add spurious entropy terms for each sensitive attribute (e.g., age, gender, race).
> - Joint treatment: Define a joint sensitive attribute over the product space (e.g., age x gender x race) and apply Eq. 3 directly to this joint attribute.
>
> FairTPT is a test-time debiasing approach, allowing the fairness auditor (user) to specify the sensitive attribute(s) at test time. For an unlabeled test image, the user can choose the attribute(s) and their possible values for debiasing. If the user specifies gender, the method is designed to debias with respect to gender only. It does not guarantee debiasing for other attributes (e.g., race). FairTPT is not intended to automatically identify spurious attributes. However, if the user knows or can infer the actual spurious factors (e.g., via external tools like GPT), these can be specified at test time.
>
> ---
>
> **Q2: Lack of explainability analysis.**
>
> Given that only the soft-prompt (context) is tuned as opposed to the Vision encoder, we report the following metrics before and after applying FairTPT (as an explanation for our method):
> - Average Sensitive Indifference (ASI): average normalized entropy of sensitive attribute predictions over the dataset (higher = less reliance on sensitive attribute).
> - Average Target Confidence (ATC): 1 - average normalized entropy of target attribute predictions over the dataset (higher = more confident target prediction).
>
> ASI table:
>
> | Dataset | Before any update | After FairTPT | After FairTPT (MO) | After TPT |
> | ----- | --- | --- | --- | --- |
> | FairFace (Gender x Race) | $42.1$ | $53.9$ | $57.4$ | $39.5$ |
> | CelebA (Hair color x Gender) | $15.3$ | $41.4$ | $46.4$ | $14.3$ |
> | CelebA (Smiling x Gender) | $15.3$ | $40.1$ | $50.9$ | $11.9$ |
> | WaterBirds (Type x Background) | $56.0$ | $70.9$ | $71.1$ | $51.2$ |
> | UTKFace (Age x Race) | $45.4$ | $53.3$ | $56.1$ | $35.3$ |
> | UTKFace (Gender x Race) | $45.4$ | $55.6$ | $57.2$ | $29.9$ |
> | Average Results | $36.6$ | $52.5$ | $56.5$ | $30.3$ |
> |
>
> ATC table:
>
> | Dataset | Before any update | After FairTPT | After FairTPT (MO) | After TPT |
> | ----- | --- | --- | --- | --- |
> | FairFace (Gender x Race) | $66.4$ | $65.0$ | $64.4$ | $88.4$ |
> | CelebA (Hair color x Gender) | $57.5$ | $53.9$ | $55.1$ | $96.0$ |
> | CelebA (Smiling x Gender) | $45.2$ | $44.2$ | $43.4$ | $80.2$ |
> | WaterBirds (Type x Background) | $62.1$ | $59.7$ | $60.7$ | $80.0$ |
> | UTKFace (Age x Race) | $46.1$ | $45.2$ | $44.8$ | $85.0$ |
> | UTKFace (Gender x Race) | $71.6$ | $70.6$ | $69.3$ | $91.4$ |
> | Average Results | $58.1$ | $56.4$ | $56.3$ | $86.8$ |
> |
>
> Observation: FairTPT/FairTPT(MO) significantly increases ASI (fairness) with minimal impact on ATC (accuracy).
>
> ---
>
> **Q3: No ablation study for FairTPT.**
>
> We provide a comprehensive sensitivity and ablation study in the experiments section (Lines 431-452):
> - Table 9 in Appendix A.3: Performance of FairTPT and FairTPT(MO) with and without ELRA.
> - Table 9 in Appendix A.3: Effect of varying $\lambda_{\text{fair}}$, including $\lambda_{\text{fair}} = \infty$ (equivalent to simply maximizing spurious entropy).
> - Table 10 in Appendix A.3: Comparison of different spurious loss terms (Eq. 3: S loss, Eq. 4: TS loss, Eq. 5: super TS loss).
>
> ---
>
> **Q4: Generalizability and sensitivity of $\lambda_{\text{fair}}$.**
>
> Table 9 in Appendix A.3 shows that FairTPT is robust across a wide range of $\lambda_{\text{fair}} \in [1, \infty)$ and across datasets, thanks to ELRA (automatic learning-rate adaptation). ELRA adaptively selects learning rates using unlabeled image inputs and interacts favorably with $\lambda_\text{fair}$ and $\rho$, enabling \textsc{FairTPT} variants to improve subgroup performance where possible while preventing accuracy collapse. No additional tuning is required. In contrast, OrthCali baseline is highly sensitive to $\lambda_{\text{orth}}$ (Table 8 in Appendix A.3), making it difficult to select a single value that generalizes well.
>
> ---
>
> We hope that our responses address your concerns and are helpful in improving your rating. If you have any other comments or feedback, please let us know! Thank you again for the review.

---

> > ### Comment · Reviewer_YdNt · 2025-11-20
> >
> > Thanks for the author’s response and the additional experiments. Apologies for Q1, where I missed the combination attributes in Table 1, and I noticed a placeholder in the soft prompting template, so I thought it might only infer one attribute at a time. I have no further questions regarding Q1. For Q2, in that regard (as shown in the ASI and ATC table), does it mean that no matter which sensitive attributes I choose (whether or not they reflect the true underlying spurious correlations), applying FairTPT can improve fairness in any case? For Q3, I have checked Tables 9 and 10. What I mean by ablation study here is the impact of each component and how it affects or contributes to the performance. For example, in an objective function A-B, one could remove B or remove A to show the difference. In this paper, a similar analysis could be done by setting $\lambda_{\text{fair}} = 0$. Q4 has been resolved, and I have no further questions.

---

> > > ### Author Response · Authors · 2025-11-20
> > >
> > > Thank you for the feedback and questions!
> > >
> > > Follow-up on Q2: No, FairTPT improves fairness only with respect to the chosen/specified sensitive attribute. Note that the ASI metric is computed w.r.t. this specified attribute.
> > >
> > > Follow-up on Q3: We agree with this point. In Table 9 (see the "FairTPT special cases" block, lines 874-875), we already include the ablation case with $\lambda_{\text{fair}} = 0$, which corresponds to removing the spurious term entirely. In Table 10, across the three blocks, we vary only the design choice of the spurious term; because the $\lambda_{\text{fair}} = 0$ setting yields the same values regardless of this design choice variation, the results are identical to those in Table 9. For completeness and readability, we can duplicate that row in each block of Table 10.
> > >
> > > If you have any other comments or feedback, please let us know!

---

> > > > ### Comment · Reviewer_YdNt · 2025-11-27
> > > >
> > > > Thank the authors for the further response and I will keep my current positive score.

---

> > > > > ### Author Response · Authors · 2025-11-27
> > > > >
> > > > > Thank you for the follow-up discussion and for maintaining the positive score. We appreciate your thoughtful review and constructive feedback.

---

### Official Review · Reviewer_hjYo · 2025-10-30

**Soundness:** 2
**Presentation:** 3
**Contribution:** 2
**Rating:** 4
**Confidence:** 3

**Summary:**

Briefly summarize the paper and its contributions. You can incorporate Markdown and Latex into your review. See https://openreview.net/faq.

This work tackles a common problem in large-scale vision-language models (VLMs) like CLIP: balancing predictive performance with the mitigation of spurious correlations, i.e. fairness in this article. Building upon the Test-Time Prompt Tuning (TPT) framework—which optimizes prompt embeddings by minimizing prediction entropy over augmented inputs—the authors introduce a sophisticated dual-objective approach. Specifically, they build upon this by introducing a dual objective, adding what is effectively a "reverse TPT" for spurious attributes. While the TPT objective minimizes entropy to improve accuracy on the target task, this new, opposing objective simultaneously maximizes the entropy for spurious features. This strategy is designed to actively "unlearn" or eliminate these biases at test time, thereby achieving enhanced fairness while preserving the model's overall performance.

**Strengths:**

A substantive assessment of the strengths of the paper, touching on each of the following dimensions: originality, quality, clarity, and significance. We encourage reviewers to be broad in their definitions of originality and significance. For example, originality may arise from a new definition or problem formulation, creative combinations of existing ideas, application to a new domain, or removing limitations from prior results.

They address a meaningful challenge of mitigating spurious correlations in VLMs without largely sacrificing accuracy, a critical issue for fair and robust deployment. Its originality lies in the clever extension of Test-Time Prompt Tuning (TPT) into a dual-objective framework. The introduction of a "reverse TPT" to actively maximize entropy for spurious attributes is an intuitively effective method for test-time bias disentanglement. Furthermore, the clarity of the paper is great as for me, presenting its "push-pull" logic and technical design with precision.

**Weaknesses:**

A substantive assessment of the weaknesses of the paper. Focus on constructive and actionable insights on how the work could improve towards its stated goals. Be specific, avoid generic remarks. For example, if you believe the contribution lacks novelty, provide references and an explanation as evidence; if you believe experiments are insufficient, explain why and exactly what is missing, etc.

The primary weakness lies in its perceived lack of substantial novelty as for me. The core algorithm is somewhat an extension of the existing Test-Time Prompt Tuning (TPT) framework. The objective is an intuitive application of TPT's entropy-based mechanism rather than a new paradigm. As such, the overall novelty of the technical contribution is somewhat limited as for me.

**Questions:**

For the final methods FAIRTPT and FAIRTPT (MO), does the latter refer to the algorithm + Multi-Objective Optimization engineering technique? And your Equation 3 and Equation 4 are actually two different designs, but for these two algorithms, did you default to choosing Equation 4 since I didn't find the specifies. I want to know what effect of Equation 3 is. Have you ever done related experiments to try it?

---

> ### Author Response · Authors · 2025-11-19
>
> We thank the reviewer for the thoughtful comments. Below, we address the concerns raised.
>
> ---
>
> **Q1: ... perceived lack of substantial novelty ... the objective is an intuitive application of TPT's entropy-based mechanism ...**
>
> We respectfully disagree with the reviewer's assessment. The simplicity or intuitive nature of a method should not be conflated with a lack of novelty. Notably, TPT  (Shu et al., 2022) itself is a simple extension of Memo (Zhang et al., 2022) for VLMs, and Zero (Farina et al., 2024) is a simple modification of TPT and Memo. Despite their simplicity, these methods were considered valuable contributions due to the meaningful empirical insights they provided.
>
> To the best of our knowledge, the fairness (subgroup robustness) implications of episodic TTA methods have not been benchmarked before, and our work is the first to introduce a fairness-aware episodic TTA method. This constitutes a new and unexplored direction within the TTA literature. Furthermore, prior studies (Zhao et al., 2023; Sheng et al., 2025) have highlighted that episodic TTA methods suffer from instability and hyperparameter sensitivity. Our method, FairTPT, directly addresses these limitations through an automatic learning-rate adaptation (ELRA) mechanism. We believe the novelty and contributions of FairTPT should be evaluated holistically, taking these factors into account.
>
> ---
>
> **Q2a: Does FairTPT (MO) mean FairTPT with Multi-Objective Optimization?**
>
> Yes.
>
> **Q2b: Are Eqs. (3) and (4) two different designs of FairTPT and FairTPT (MO)? Did you default to Eq. (4) because I didn't see specifics?**
>
> Equations 3 (S loss), 4 (TS loss), and 5 (super TS loss) are design variants for both FairTPT and FairTPT (MO). As noted in Section 5 (lines 354-355), we use Eq. (3) (S loss) as the default in main experiments.
>
> **Q2c: What effect does Eq. (3) have? Did you test it?**
>
> Table 10 in Appendix A.3 compares all three variants for both FairTPT and FairTPT (MO). We found no significant gains from Eqs. (4) or (5) over Eq. (3), supporting our choice of the simpler S loss as default.
>
> ---
>
> We hope that our responses address your concerns and are helpful in improving your rating. If you have any other comments or feedback, please let us know! Thank you again for the review.

---

> > ### Comment · Reviewer_hjYo · 2025-11-20
> >
> > Thank you for your reply and for referencing the comparisons in Appendix A.3.
> >
> > I have reviewed Table 10 and noticed that the default Equation (3) appears to be the most effective setting. This raises a question regarding the design motivation: Equations (4) and (5) were introduced in the paper as methodological improvements over Eq. (3). However, since they are extensions of Eq. (3), one would expect them to yield better performance.
> >
> > The current results suggest that the added complexity in Eqs. (4) and (5) do not lead to the intended benefits, which seems to challenge the initial viewpoints presented for their design. Could you offer some additional explanation or analysis as to why these variants fell short of expectations compared to the simpler S loss?

---

> > > ### Author Response · Authors · 2025-11-20
> > >
> > > Thank you for the question. We clarify that Eqs. (4) and (5) were introduced not as methodological improvements over Eq. (3), but as alternative and intuitive design choices for the spurious term. We intended to provide a thorough empirical comparison of these plausible variants.
> > >
> > > Although Eqs. (4) and (5) offer more flexibility in reducing reliance on the sensitive attribute; this flexibility can also introduce trade-offs: depending on the dataset-attribute configuration, these variants may slightly reduce overall accuracy, which in turn affects fairness metrics. Our learning rate adaptation (ELRA) strategy helps counteract potential accuracy collapse in such cases.
> > >
> > > Consequently, the relative effectiveness of Eqs. (4) and (5) is dataset-dependent. In Table 12 (last three blocks), we report results for additional dataset-attribute configurations, where Eq. (5) appears to be more effective. Overall, the three formulations exhibit broadly comparable performance (in terms of average results, consistency across datasets, and robustness to $\lambda_\text{fair}$), so we advocate the simplest choice (Eq. 3).
> > >
> > > If you have any other comments or feedback, please let us know!

---

> > > > ### Author Response · Authors · 2025-11-28
> > > >
> > > > Dear Reviewer,
> > > >
> > > > Thank you once again for your valuable feedback and follow-up discussion. As the discussion deadline approaches, we kindly request you to review our response to your follow-up question. We are happy to address any remaining concerns. Thank you for your time and effort!
> > > >
> > > > Regards,
> > > > Authors

---

### Official Review · Reviewer_jwNW · 2025-11-01

**Soundness:** 2
**Presentation:** 3
**Contribution:** 1
**Rating:** 4
**Confidence:** 3

**Summary:**

This paper proposes a method which aims at test-time tuning of zero-shot image classification tasks in visual-language models. The authors introduce two loss terms associated with the prompt generation. The authors evaluate their method against a zero-shot baseline,fine-tuned model, and a debiasing baseline, on 4 datasets.

**Strengths:**

1. Overall, this paper is well-motivated and presents the technical details well. It would be suitable for a general ai researcher outside the area.

2. The evaluation is suitable thorough, the datasets and baselines are reasonable. Had there been a stronger delta against the baselines, this would be a complete paper.

3. The authors give detailed outline of all method pseudocode (but no included code) in the appendix. Also significant secondary results in the appendix. This is therefore an acceptable (but not exceptional) standard of reproducibility.

**Weaknesses:**

1. Overall, the evaluation is underwhelming. As is common in debiasing work, the authors don't provide a qualitative evaluation on a downstream application. For example, is +2.0% WGA qualitatively different than +1.1 WGA when comparing OrthCali v. FairTPT on average performance (Table 1)? Further, the authors don't provide variance over the trials (more than 5 trials would be best to estimate variance).

2. The results against the fairness baseline are similar enough that things like training or inference cost are relevant, e.g. is the delta improvement over the baseline due to better hyperparameter tuning, over a larger set of candidate models or higher training runtime? Is there a large difference in inference-time cost that would justify a small delta improvement?

3. The results show significant parameter sensitivity on η. This is concerning, as this may increase training time on arbitrary datasets where we can't assume prior hyperparameters are suitable.

Together, the authors haven't demonstrated a significant improvement in terms of efficiency or qualitative results.

**Questions:**

From above weaknesses:

1. What is the qualitative value of these delta improvements observed against the fair baseline?

2. Do you have std results that can be reported? At least in the average columns (if readability is a concern)?

3. Do you have runtime comparisons, both in terms of training/tuning time, and inference?

---

> ### Author Response · Authors · 2025-11-19
> **Author Response (part-1)**
>
> We thank the reviewer for the thoughtful comments. Below, we address the concerns raised.
>
> ---
>
> **Q1: Delta improvements against baselines and the qualitative significance of those improvements.**
>
> Regarding deltas against the zero-shot and episodic TTA baselines: We would like to highlight that the magnitude of improvements we report is in line with prior episodic TTA works. For example, Tables 1 and 2 in Zero (Farina et al., 2024) report only ~0-4\% overall accuracy gains (for both TPT and Zero) over the zero-shot baseline across several datasets. In this context, the fairness-oriented improvements (delta) of FairTPT over zero-shot and episodic TTA baselines across the dataset–attribute configurations in our work (Tables 1 and 12) are reasonable and consistent with the broader TTA literature. Importantly, prior episodic TTA works do not evaluate on fairness-sensitive datasets.
>
> Regarding deltas against the OrthCali baseline: OrthCali is a strong test-time debiasing method, but the merits of FairTPT should not be judged solely by average performance across datasets. Two additional factors are important:
> - Consistency across datasets: In Table 1, OrthCali does not consistently outperform the zero-shot baseline across datasets, whereas FairTPT consistently shows improvements under a fixed hyperparameter choice.
> - Robustness to hyperparameters: In unsupervised test-time adaptation, it is difficult to identify dataset-specific hyperparameters using only unlabeled data. As shown in Table 8, OrthCali is highly sensitive to the choice of $\lambda_{\text{orth}}$, making it challenging to select a single value that performs well across datasets. In contrast, Table 9 shows that FairTPT remains robust across a broad range of $\lambda_{\text{fair}} \in [1, \infty)$.
> Taking these considerations together, we view FairTPT as substantially stronger and more reliable than the OrthCali baseline.
>
> Regarding qualitative significance of improvements (e.g., WGA): Qualitative interpretation of fairness metric improvements is inherently application-dependent, requiring domain-specific socio-technical considerations. If the reviewer has a specific qualitative protocol in mind (or a reference to such a qualitative evaluation based on algorithmic fairness metrics), we would be happy to incorporate a discussion in the final draft.
>
> ---
>
> **Q2: Small delta improvements against the fairness baseline (OrthCali) ... is the delta improvement over the baseline due to better hyperparameter tuning ... runtime comparisons of these methods**
>
> As noted above, the comparison to OrthCali should not rely solely on average deltas but also consider consistency across datasets and hyperparameter robustness.
>
> Hyperparameter tuning: The improvements of FairTPT over OrthCali are not a result of better hyperparameter tuning. FairTPT is robust to $\lambda_{\text{fair}}$, whereas OrthCali is highly sensitive to $\lambda_{\text{orth}}$.
>
> Runtime: FairTPT's inference-time cost is comparable to the TPT baseline. Below we report the runtime per image (sec) on the FairFace (Gender x Race) configuration:
>
> TPT: 0.43 | Zero:  0.04 | FairTPT: 1.03 | FairTPT (MO): 1.71
>
> OrthCali only requires solving an optimization problem upfront, and its inference-time cost is similar to the zero-shot baseline. The reported time for FairTPT already includes its automatic learning-rate adaptation (ELRA). Because FairTPT is robust to $\lambda_{\text{fair}}$, no additional tuning is needed. In contrast, OrthCali would require tuning $\lambda_{\text{orth}}$, and such tuning would require a labeled validation set (contrary to the unsupervised test-time setting).
>
> ---
>
> **Q3: Sensitivity to learning rate $\eta$.**
>
> Only the TPT baseline is sensitive to $\eta$. FairTPT addresses this by employing an automatic learning-rate adaptation (ELRA) strategy. Both TPT and FairTPT perform only a single-epoch update using augmentations of the test image, but FairTPT efficiently identifies a suitable learning rate with only a few restarts. The computational overhead is comparable to TPT.

---

> > ### Author Response · Authors · 2025-11-19
> > **Author Response (part-2)**
> >
> > ---
> >
> > **Q4: Reproducibility of the work.**
> >
> > We have included full pseudocode for all compared methods and all hyperparameter settings in the appendix. We will release the full code as part of the final version.
> >
> > ---
> >
> > **Q5: Do you have std results that can be reported?**
> >
> > We reported mean values to maintain table readability, as the variance is low. Below, we provide the mean $\pm$ std values for the FairFace (Gender x Race) column in Table 1:
> >
> > | Method | Acc | WGA | Bias | EOD |
> > | ----- | --- | --- | --- | --- |
> > | Zero-Shot | $95.7 \pm 0.06$ | $90.0 \pm 0.15$ | $5.7 \pm 0.09$ | $9.4 \pm 0.12$ |
> > | TPT | $93.6 \pm 0.09$ | $85.0 \pm 0.28$ | $8.6 \pm 0.24$ | $12.4 \pm 0.23$ |
> > | Zero | $91.2 \pm 0.05$ | $78.6 \pm 0.31$ | $12.6 \pm 0.33$ | $11.9 \pm 0.39$ |
> > | OrthCali | $95.9 \pm 0.09$ | $87.8 \pm 0.29$ | $8.1 \pm 0.21$ | $11.3 \pm 0.29$ |
> > | FairTPT | $95.5 \pm 0.08$ | $90.7 \pm 0.15$ | $4.8 \pm 0.09$ | $8.1 \pm 0.11$ |
> > | FairTPT (MO) | $95.3 \pm 0.09$ | $90.4 \pm 0.15$ | $4.9 \pm 0.11$ | $7.9 \pm 0.16$ |
> > |
> >
> > For reference, TPT  (Shu et al., 2022) and Zero (Farina et al., 2024) report averages over 3 seeds; we report results over 5 seeds. We will include mean $\pm$ std for all the columns in Table 1 in the final code repository.
> >
> > ---
> >
> > We hope that our responses address your concerns and are helpful in improving your rating. If you have any other comments or feedback, please let us know! Thank you again for the review.

---

> > > ### Author Response · Authors · 2025-11-28
> > >
> > > Dear Reviewer,
> > >
> > > Thank you once again for your valuable feedback. As the discussion deadline approaches, we kindly request you to review our rebuttal, where we have addressed your concerns to the best of our ability. We are happy to address any remaining concerns. Thank you for your time and effort!
> > >
> > > Regards,
> > > Authors

---

### Official Review · Reviewer_1PZi · 2025-11-03

**Soundness:** 3
**Presentation:** 2
**Contribution:** 3
**Rating:** 6
**Confidence:** 2

**Summary:**

The paper evaluates the fairness of existing test-time adaptation (TTA) methods for VLMs and proposes a new TTA approach to reduce spurious correlations. First, the paper shows that the existing TTA methods do not improve subgroup robustness, can amplify disparities, and are highly sensitive to hyperparameters. Then, a new TTA method, FAIRTPT, is proposed that jointly minimizes the target-attribute entropy to preserve accuracy and maximizes the sensitive-attribute entropy to reduce the spurious correlations. Empirically, FAIRTPT achieves SOTA/competitive results across different fairness benchmarks.

**Strengths:**

- The paper studies an important problem and proposes a rigorous solution.

- Empirically, FAIRTPT outperforms other TTA methods across various fairness benchmarks.

**Weaknesses:**

- I find the hyperparameter sensitivy aspect a bit irrelevant to the main theme of the paper, which is fairness. While it is nice that FAIRTPT is a method with more favorable hyperparameter sensitivy than other TTA methods, a presentation where this is introduced as a nice-to-have property would make the overall story more coherent.

- The method is specifically developed and empirically validated on zero-shot classification asks, but the title sounds more general. This specific focus/limitation should be more explicit in the title.

**Questions:**

-

---

> ### Author Response · Authors · 2025-11-19
>
> We thank the reviewer for the thoughtful comments. Below, we address the concerns raised.
>
> ---
>
> **Q1: ... the hyperparameter sensitivity aspect is a bit irrelevant to the main theme of the paper ... presenting it as a nice-to-have property would make the narrative more coherent ...**
>
> The main theme of our work is fairness in test-time adaptation (TTA) methods. Prior work has highlighted instability and hyperparameter sensitivity in TTA methods (Zhao et al., 2023; Sheng et al., 2025), which raises serious concerns for fairness-critical settings. When hyperparameters vary, subgroup performance can fluctuate dramatically, often amplifying disparities rather than reducing them. For this reason, we view hyperparameter robustness as an essential property for fairness-aware TTA methods to enable reliable deployment.
>
> In the experiments section, we noted:
> "Across datasets, episodic TTA baselines are not reliably accuracy-improving and often harm subgroup performance unless hyperparameters are carefully chosen, something that is particularly hard at test time with only unlabeled inputs."
> By explicitly addressing this issue, FairTPT reduces the risk of fairness degradation caused by hyperparameter sensitivity.
>
> ---
>
> **Q2: The method is developed and validated for zero-shot classification ... it should be made explicit in the title.**
>
> Similar to prior episodic TTA works (e.g., TPT and Zero), our focus is on zero-shot classification tasks. As suggested, we could update the title to: "Fairness-Aware Episodic Test-Time Prompt Tuning" or "Fairness-Aware Test-Time Prompt Tuning for Zero-Shot Classification".
>
> ---
>
> We hope that our responses address your concerns and are helpful in improving your rating. If you have any other comments or feedback, please let us know! Thank you again for the review.

---

> > ### Comment · Reviewer_1PZi · 2025-11-25
> >
> > I thank the authors for addressing some comments!

---

> > > ### Author Response · Authors · 2025-11-27
> > >
> > > Thank you for the response. If you have any remaining comments, please let us know!

---

### Author Response · Authors · 2025-11-28
**Rebuttal Summary**

We thank all reviewers for their thoughtful feedback. Below is a summary of our rebuttal points:

To the best of our knowledge, the fairness (subgroup robustness) implications of episodic TTA methods have not been benchmarked before, and our work is the first to introduce a fairness-aware episodic TTA method. Furthermore, prior studies (Zhao et al., 2023; Sheng et al., 2025) have noted that episodic TTA methods suffer from instability and hyperparameter sensitivity. Our method, FairTPT, directly addresses these limitations through an automatic learning-rate adaptation (ELRA) mechanism.

**Key merits of FairTPT extend beyond average performance across datasets:**
- Consistency across datasets: As shown in Table 1, OrthCali does not consistently outperform the zero-shot baseline, whereas FairTPT achieves consistent improvements under a fixed hyperparameter choice.
- Robustness to hyperparameters: In unsupervised TTA, identifying dataset-specific hyperparameters using only unlabeled data is challenging. Table 8 shows OrthCali's high sensitivity to $\lambda_{\text{orth}}$, making it difficult to select a single value that works well across datasets. In contrast, Table 9 demonstrates that FairTPT remains robust across a broad range of $\lambda_{\text{fair}} \in [1, \infty)$.

**Comprehensive sensitivity and ablation studies are provided:**
- Table 9 in Appendix A.3: Performance of FairTPT and FairTPT(MO) with and without ELRA.
- Table 9 in Appendix A.3: Effect of varying $\lambda_{\text{fair}}$, including $\lambda_{\text{fair}} = \infty$ (equivalent to simply maximizing spurious entropy).
- Table 10 in Appendix A.3: Comparison of different spurious loss terms (Eq. 3: S loss, Eq. 4: TS loss, Eq. 5: super TS loss).

**Additional validation during rebuttal:**

We report two metrics before and after applying FairTPT:
- Average Sensitive Indifference (ASI): average normalized entropy of sensitive attribute predictions over the dataset (higher = less reliance on sensitive attribute).
- Average Target Confidence (ATC): 1 - average normalized entropy of target attribute predictions over the dataset (higher = more confident target prediction).

FairTPT/FairTPT(MO) significantly improves ASI (fairness) with minimal impact on ATC (accuracy).

ASI table:

| Dataset | Before any update | After FairTPT | After FairTPT (MO) | After TPT |
| ----- | --- | --- | --- | --- |
| FairFace (Gender x Race) | $42.1$ | $53.9$ | $57.4$ | $39.5$ |
| CelebA (Hair color x Gender) | $15.3$ | $41.4$ | $46.4$ | $14.3$ |
| CelebA (Smiling x Gender) | $15.3$ | $40.1$ | $50.9$ | $11.9$ |
| WaterBirds (Type x Background) | $56.0$ | $70.9$ | $71.1$ | $51.2$ |
| UTKFace (Age x Race) | $45.4$ | $53.3$ | $56.1$ | $35.3$ |
| UTKFace (Gender x Race) | $45.4$ | $55.6$ | $57.2$ | $29.9$ |
| Average Results | $36.6$ | $52.5$ | $56.5$ | $30.3$ |
|

ATC table:

| Dataset | Before any update | After FairTPT | After FairTPT (MO) | After TPT |
| ----- | --- | --- | --- | --- |
| FairFace (Gender x Race) | $66.4$ | $65.0$ | $64.4$ | $88.4$ |
| CelebA (Hair color x Gender) | $57.5$ | $53.9$ | $55.1$ | $96.0$ |
| CelebA (Smiling x Gender) | $45.2$ | $44.2$ | $43.4$ | $80.2$ |
| WaterBirds (Type x Background) | $62.1$ | $59.7$ | $60.7$ | $80.0$ |
| UTKFace (Age x Race) | $46.1$ | $45.2$ | $44.8$ | $85.0$ |
| UTKFace (Gender x Race) | $71.6$ | $70.6$ | $69.3$ | $91.4$ |
| Average Results | $58.1$ | $56.4$ | $56.3$ | $86.8$ |
|

We have uploaded a revised PDF (Appendix A.4) incorporating new discussion points from the rebuttal period. We hope these clarifications address your concerns and assist in your evaluation. Thank you again for your thoughtful review.

---

### Meta-Review · Area_Chair_oovx · 2025-12-18

**Summary:**

While the reviewers agree the paper solves the critical problem of fair and robust deployment by extending TPT into a dual-objective framework, there are also concerns on the novelty, technical details, evaluation, analysis, sensitivity, and presentation. While the authors made significant efforts to address many of these concerns, some of them seem to remain based on the discussions. In addition, two reviewers are still on the negative side, while the other two are positive, but one is not confident. Overall, despite some strong points, there is not enough support to accept this paper.

**Reviewer Concerns:**

There are concerns on the novelty, technical details, evaluation, analysis, parameter sensitivity, and presentation. The authors addressed most of them, but based on the discussions, there seems to be remaining concerns on the evaluation and parameter sensitivity (Reviewer jwNW) and analysis (Reviewer hjYo).

**Reviewer Scores:**

* Reviewer 1PZi: score is 6, but confidence is 2
* Reviewer jwNW: score is 4
* Reviewer hjYo: score is 4 even after dicussion
* Reviewer YdNt: score is 6

The AC does not see these scores changing further.

---

### Decision · Program_Chairs · 2026-01-26

Reject